# LANGUAGE MODELS ARE IMPLICITLY CONTINUOUS

**Samuele Marro**[1][*][†]    **Davide Evangelista**[2][*]    **X. Angelo Huang**[3]    **Emanuele La Malfa**[4]

**Michele Lombardi**[2]                                                        **Michael Wooldridge**[4]

[1]Department of Engineering Science
 University of Oxford
 Oxford, UK

[2]Department of Computer Science
 University of Bologna
 Bologna, Italy

[3]Department of Computer Science
 ETH Zurich
 Zurich, Switzerland

[4]Department of Computer Science
 University of Oxford
 Oxford, UK

## ABSTRACT

Language is typically modelled with discrete sequences. However, the most successful approaches to language modelling, namely neural networks, are continuous and smooth function approximators. In this work, we show that Transformer-based language models implicitly learn to represent sentences as continuous-time functions defined over a continuous input space. This phenomenon occurs in most state-of-the-art Large Language Models (LLMs), including Llama2, Llama3, Phi3, Gemma, Gemma2, and Mistral, and suggests that LLMs *reason* about language in ways that fundamentally differ from humans. Our work formally extends Transformers to capture the nuances of time and space continuity in both input and output space. Our results challenge the traditional interpretation of how LLMs understand language, with several linguistic and engineering implications.

## 1 INTRODUCTION

In linguistics and computer science, language is typically modelled as a discrete sequence of symbols: a sentence is a sequence of words, phonemes, characters, or tokens drawn from a finite vocabulary. This characterisation underpins both linguistics (Hockett & Hockett, 1960; Chomsky, 1995; Studdert-Kennedy, 2005; Akmajian et al., 2017) as well as classic and recent algorithmic approaches to language modelling (Manning, 1999; Bengio et al., 2000; Mnih & Hinton, 2008).[1] In Machine Learning, a successful paradigm to model language is that of Large Language Models (LLMs, (Devlin, 2018; Brown et al., 2020)). In LLMs, language is modelled via an optimisation problem whose objective is to predict a word given its surrounding context (Peters et al., 2018; Radford et al., 2019), though recent advancements fine-tune the models with procedures inspired by reinforcement learning (Schulman et al., 2017; Rafailov et al., 2024).

At their core, the architectures that model language, including feed-forward neural networks (Mikolov et al., 2013a), Long-Short Term Memory Networks (LSTMs) (Hochreiter, 1997; Sundermeyer et al., 2012) and Transformers (Vaswani et al., 2017), approximate a discrete sequence of tokens with continuous smooth functions. However, **training inherently continuous models on discrete sequences does not imply that the models themselves treat language as discrete**.

This paper explores how the tension between discrete data and continuous function approximators is synthesised in Transformers-based Large Language Models (Vaswani et al., 2017). To do

---

[*]These authors contributed equally.

[†]Corresponding author. Email: `samuele.marro@eng.ox.ac.uk.`.

[1]For completeness, a few notable works in linguistics and computer science model language as continuous: among the others, Alkhouli et al. (2014) and Bowman et al. (2015) model sentences as continuous entities in latent space, while recent approaches with quantum NLP represent meaning as a superstate of different words (Guarasci et al., 2022).

so, we seamlessly generalise the Transformers architecture to support continuous inputs. This extension, which does not modify a model's weights or alter the architecture, allows the study of existing pretrained LLMs, including Llama (Dubey et al., 2024), Mistral (Jiang et al., 2023), and Gemma (Gemma Team et al., 2024b), with continuous input sequences.

By running experiments on state-of-the-art LLMs, we find that the language LLMs learn is implicitly continuous, as they are able to handle, with minor modifications, inputs that are both *time continuous* and *spatial continuous*. In particular, we formally show that the results obtained by extending pretrained LLMs to handle time continuous input strongly depend on a quantity, named *duration*, associated with each sentence. We also show in Section 4 that the semantics of this *continuum* significantly deviate from human intuition. Our results suggest that our intuition about human language can be misleading when applied to LLMs, as LLMs hold implicit representations of continuous sequences in unintuitive ways. Furthermore, these observations have practical consequences from an engineering perspective, as they suggest that it is possible to leverage the continuous representations of LLMs to pretrain them more efficiently.

Our code is available at https://github.com/samuelemarro/continuous-llm-experiments.

## 2 RELATED WORK

Modern state-of-the-art pretrained language models operate on a discrete number of tokens and do not handle continuous inputs directly. However, in other domains, extensions of classical Transformers (Vaswani et al. (2017)) to time continuous inputs have been recently explored in tackling different problems. In modelling dynamical systems, (Fonseca et al. (2023)) have proposed adding new regularisations to a classical transformer to create continuous behaviour in an attempt to improve upon existing time continuous models such as Neural ODEs (Chen et al. (2019); Kidger (2022)). In time series modelling, (Chen et al. (2024); Moreno-Pino et al. (2024)) further developed the ideas advanced by Neural ODEs by integrating time continuous transformers and, consequently, superseding other existing approaches such as Neural CDE (Kidger et al. (2020)) and Neural RDE (Morrill et al. (2021)).

Another line of work considers time continuous extension of language by processing it through networks combining the flexibility of classical Transformers with the mathematical interpretability of NeuralODEs, such as *ODETransformer* (Li et al. (2022a)), *CSAODE* (Zhang et al. (2021)), *TransEvolve* (Dutta et al. (2021)), and *N-ODE Transformer* (Baier-Reinio & De Sterck (2020)). Several authors have also explored spatial continuous extensions of LLMs Tang et al. (2021); Schwenk (2007); Östling & Tiedemann (2017), where the embedding space is expanded to include vectors not directly mapped to specific tokens, thereby enhancing the representational power of the models. A broad class of Diffusion Language Models (Li et al. (2022b); Gong et al. (2022); Lovelace et al. (2024a); Gulrajani & Hashimoto (2024); Zhang et al. (2024); Lovelace et al. (2024b)) employs a similar concept, where the model generates elements not necessarily tied to individual tokens in the embedding space, thereby effectively incorporating spatial continuous extensions into language modelling.

Additionally, continuous representations in LLMs have been studied either in the context of concepts for which intuitive spectra exist (Gurnee & Tegmark, 2023; Arditi et al., 2024) or from a neuron-driven perspective (Anthropic, 2024). In particular, our work can be seen as complementary to the latter: while the authors of Anthropic (2024) show that certain neurons map to specific concepts (which can then be steered), we show that such concepts exist even at the embedding level, and we offer a theoretical framework to study formally such phenomena in an architecture-independent manner.

Continuing the overview of related works, in Ravfogel et al. (2022) the authors remove concepts like biases by erasing the pertinent area that makes that concept emerge. Our work shows however that LLMs can interpolate between "overlapping" concepts: erasing an entire area might thus also affect other concepts, which represents an interesting research direction. Moreover, in Todd et al. (2023) the authors show that some LLMs feature *function vectors*, which represent specific operations (e.g. sums), It is possible that some of the continuous behaviours in an LLM arise as function vectors.

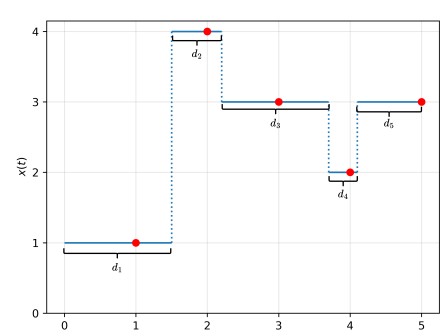 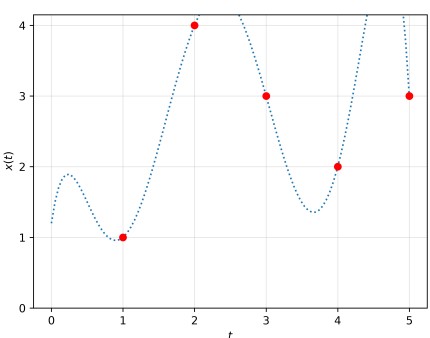

Figure 1: *Left.* A graphical representation of the time continuity of language. Each observed token is obtained by sampling at integer timesteps a stepwise constant function defined on the real interval $[0, T]$. The length of each constant interval is the duration of the associated token. *Right.* A spatial continuous extension of a sentence, where $\mathbf{x}(t)$ can represent any value.

Finally, our work can also be seen as a response to Vilnis & McCallum (2014), which trains inherently continuous embeddings: we show that this process is not necessary, as even discretely trained LLMs show similar behaviours.

## 3  A CONTINUOUS GENERALISATION OF LANGUAGE MODELS

In this section, we propose the hypothesis that language can be seen as the discretisation of a spatio-temporal continuous function whose value corresponds to a valid token for any integer timestep. As we will show, this assumption allows us to define a continuous extension of the classical casual Transformer module, namely the Continuous Causal Transformer (CCT). The CCT accepts spatio-temporal continuous functions as input while including pretrained models on regular time- and space-discrete data as a special case. Moreover, we will formally discuss the implications of this construction, showing the basic results required to describe the experiments we presented later.

### 3.1  TIME CONTINUITY

Following classical approaches (Cotterell et al., 2023), we define a natural sentence as a sequence $\{\mathbf{w}_1, \mathbf{w}_2, \dots, \mathbf{w}_T\} \subset \mathcal{W}$ of tokens, sampled from an underlying distribution $p(\mathbf{w}_1, \mathbf{w}_2, \dots, \mathbf{w}_T)$, where each token $\mathbf{w}_t$ only depends on previous timesteps, i.e. $p(\mathbf{w}_1, \mathbf{w}_2, \dots, \mathbf{w}_T) = p_1(\mathbf{w}_1) \prod_{t=2}^{T} p_t(\mathbf{w}_t | \mathbf{w}_{<t})$, where we defined $\mathbf{w}_{<t} := (\mathbf{w}_1, \dots, \mathbf{w}_{t-1})$. Moreover, given a continuous function $\mathcal{E} : \mathcal{W} \to \mathbb{R}^d$ that *embeds* any token $\mathbf{w}_t$ as a $d$-dimensional vector $\mathbf{x}_t := \mathcal{E}(\mathbf{w}_t)$, we name the push-forward distribution $p(\mathbf{x}_{\leq T}) := \mathcal{E}_{\#} p(\mathbf{w}_{\leq T})$ the *distribution of natural sentences*. Clearly, since the set $\mathcal{W}$ is inherently discrete, then the set $\mathcal{X} = Rg(\mathcal{E})$, i.e., the range of $\mathcal{E}$, is a finite set, to which we refer as the *space of valid embeddings*. Indeed, in any classical formalisation of language, a sentence is considered to be finite both in time and space.

In this work, we hypothesise that any observed sentence $\{\mathbf{x}_t\}_{t=1}^{T} \sim p(\mathbf{x}_{\leq T})$ originates from the integer discretisation of a function $\mathbf{x}(t)$, defined on the real interval $[0, T]$. Clearly, there exist infinitely many of these functions. In this section, we assume for $\mathbf{x}(t)$ the simplest, possible form: a stepwise constant function, defined as

$$\mathbf{x}(t) = \sum_{s=1}^{T} \mathbf{x}_s \mathbb{1}_{[a_s, b_s]}(t), \tag{1}$$

where $\mathbb{1}_{[a_s, b_s]}(t)$ is the indicator function of the interval $[a_s, b_s]$, whose value is 1 if $t \in [a_s, b_s]$, 0 otherwise. For the intervals $\{[a_s, b_s]\}_{s=1}^{T}$ we assume that:

($\mathcal{H}1$)  $\{[a_s, b_s]\}_{s=1}^{T}$ define a partition of the interval $[0, T]$, i.e.

$$\bigcup_{s=1}^{T} [a_s, b_s] = [0, T], \qquad \bigcap_{s=1}^{T} [a_s, b_s] = \varnothing.$$

($\mathcal{H}2$)  $s \in [a_s, b_s]$ for any $s = 1, \ldots, T$.

With this assumption, a sentence can be seen as a continuous flow of information, where the *duration* of each word is defined as the length of the interval defining it, i.e. $b_s - a_s$. Throughout the rest of this paper, we will refer to the duration of a token $\mathbf{x}_s$ as $d_s := b_s - a_s$. A representation of this concept is summarised in Figure 1.

To be able to process time continuous inputs, we need an extension of the typical causal Transformer architecture, which we name *Continuous Causal Transformer (CCT)*. We argue that any Transformer-based LLM can be seen as a discretisation of our CCT, and we propose a technique to modify the architecture of standard LLMs to handle stepwise-constant sentences as input, which represents the basis of our experiments in Section 4.

To prove our statement, we begin by considering the classical formulation of multi-head causal attention module, where the transformed output $\{\mathbf{y}_t\}_{t=1}^{T}$ associated with the sequence $\{\mathbf{x}_t\}_{t=1}^{T}$ is defined as:

$$\mathbf{y}_t = \sum_{s=1}^{t} \frac{1}{Z_t} \exp\left(\frac{\mathbf{q}_t^T \mathbf{k}_s}{\sqrt{d}}\right) \mathbf{v}_s, \tag{2}$$

where $Z_t := \sum_{s'=1}^{t} \exp\left(\frac{\mathbf{q}_t^T \mathbf{k}'_s}{\sqrt{d}}\right)$ is a normalisation constant, and:

$$\mathbf{q}(t) = W^{(q)} \mathbf{x}(t), \quad \mathbf{k}(t) = W^{(k)} \mathbf{x}(t), \quad \mathbf{v}(t) = W^{(v)} \mathbf{x}(t), \tag{3}$$

where $W^{(q)}$, $W^{(k)}$, and $W^{(v)}$ are the attention matrices that are learned during training. For an in-depth derivation of Equation 2, refer to Appendix A.1.

Therefore, a continuous extension of Equation 2 can be simply obtained by substituting the sum with an integral, thus obtaining the *continuous causal attention module*

$$\mathbf{y}(t) = \int_0^t \frac{1}{Z_t} \exp\left(\frac{\mathbf{q}(t)^T \mathbf{k}(s)}{\sqrt{d}}\right) \mathbf{v}(s) ds, \tag{4}$$

where $Z_t := \int_0^t \exp\left(\frac{\mathbf{q}(t)^T \mathbf{k}(s)}{\sqrt{d}}\right) ds$.

The multi-head version of Equation 4 is obtained by simply considering $H$ independent copies of $\mathbf{y}(t)$ (namely, $\{\mathbf{y}_1(t), \ldots, \mathbf{y}_H(t)\}$), concatenating them and multiplying the result with a parameter matrix $W^{(o)}$:

$$\mathbf{y}(t) = W^{(o)} \texttt{cat}(\mathbf{y}_1(t), \ldots, \mathbf{y}_H(t)). \tag{5}$$

Note that the learned matrices $W^{(q)}, W^{(k)}, W^{(v)}$, and $W^{(o)}$ do not depend on the time discretisation (since they act on the feature domain of $\mathbf{x}_t$ and $\mathbf{y}_t$). Consequently, we can use pretrained weights from any classical LLM.

To complete the construction of our CCT architecture, we remark that the $\texttt{Add\&Norm}$ scheme can be naturally re-used as it is, computing the final output $\tilde{\mathbf{x}}(t)$ as:

$$\tilde{\mathbf{x}}(t) = \texttt{LayerNorm}(W^{(z)} \mathbf{z}(t) + \mathbf{x}(t)), \tag{6}$$

$$\mathbf{z}(t) = \texttt{LayerNorm}(\mathbf{y}(t) + \mathbf{x}(t)). \tag{7}$$

Finally, we observe that Equation 4 can be simplified by considering our stepwise-constant assumption for the language, as in Equation 1. Indeed, we show in Appendix A.2 that:

$$\mathbf{y}(t) = \sum_{k=1}^{T} \frac{1}{Z_t} \exp\left(\frac{\mathbf{q}_{\bar{t}}^T \mathbf{k}_k}{\sqrt{d}}\right) \mathbf{v}_k d_k, \tag{8}$$

where $\bar{t}$ is the only integer such that $t \in [a_{\bar{t}}, b_{\bar{t}}]$, whose existence and uniqueness are guaranteed by $(\mathcal{H}1)$ and $(\mathcal{H}2)$. Note that Equation 8 is equivalent to Equation 2 as long as the partition $\{[a_k, b_k]\}_{k=1}^{T}$ is chosen so that each token has a uniform duration $d_k = 1$ for all $k = 1, \ldots, T$, since in this case for any integer timestep $t$,

$$\mathbf{y}_t = \mathbf{y}(t) = \sum_{k=1}^{T} \frac{1}{Z_t} \exp\left(\frac{\mathbf{q}_t^T \mathbf{k}_k}{\sqrt{d}}\right) \mathbf{v}_k, \tag{9}$$

which is exactly Equation 2.

However, the CCT module is more general in the sense that it can easily handle arbitrarily stepwise constant functions, for example with non-uniform duration. Indeed, we can simply choose any partition of the interval $[0, T]$ into sub-intervals satisfying condition $(\mathcal{H}2)$, and apply Equation 8 to compute the continuous transformed output $\mathbf{y}(t)$. Interestingly, Equation 8 implies that, for a fixed input sentence $\mathbf{x}(t)$, the output $\mathbf{y}(t)$ does not depends on the chosen partition $\{[a_k, b_k]\}_{k=1}^{T}$, but only on the durations $d_k$ of each token. This suggests that our CCT shows shifting-invariance, i.e. the output does not change if the input gets shifted in time, as long as the shifted partition satisfies $(\mathcal{H}1)$ and $(\mathcal{H}2)$. On the other hand, the CCT is not scale-invariant, since scaling will alter the token duration. These properties have been observed empirically in Appendix D.1.

## 3.2 SPACE CONTINUITY

Note that, in our construction, we never explicitly used the fact that the range of $\mathbf{x}(t)$ is a subset of $\mathcal{X}$, i.e. that any value of $\mathbf{x}(t)$ is an admissible token. This motivates the introduction of spatial continuous CCTs, where we consider a more general sentence in the form of equation 1, where $\mathbf{x}_s$ is not necessarily the embedding of a meaningful word $\mathbf{w}_s$. Note that our CCT model does not require any explicit modifications to be adapted to this setup.

Spatial continuity is particularly relevant when the value $\mathbf{x}_s \notin \mathcal{X}$ is obtained by interpolating between two meaningful tokens. Interestingly, we show in Section 4.3 that pretrained LLMs assign a semantic meaning to these intermediate embeddings, which results in something distinct from the two tokens which are used to compute the interpolation. Note that this is non-trivial and non-predictable, since the LLM never explicitly saw the majority of non-meaningful tokens as input during training, suggesting intriguing properties of the CCT architecture itself (which we aim to analyse more in-depth in future work).

To conclude, we remark that the stepwise constant assumption on the continuous language structure as in Equation 1 can be easily generalised to any other function structure for which a closed-form solution of the integral in Equation 4 can be obtained. This happens, for example, for any choice of piece-wise polynomial function defined over a partition of $[0, T]$, satisfying the assumptions $(\mathcal{H}1)$ and $(\mathcal{H}2)$. Testing the behaviour of the CCT architecture for other choices of $\mathbf{x}(t)$ is left to future work.

In the next sections, we will prove through several experiments that not only our proposed CCT model acts as a *natural generalisation* of classical Causal Transformer, recovering the same behaviour when a uniform discretisation of the domain into intervals of length 1 is considered, but also that the CCT seems to understand the concept of *temporal fraction of a word*.

## 4 PRETRAINED LLMS ARE IMPLICITLY CONTINUOUS

In this section, we use our generalisation of LLMs to study the time continuous and space-continuous behaviours of pretrained LLMs. Thanks to our extension, we identify several novel properties of dis-

cretely trained models, such as the key role of duration as a semantic component and the existence of meaningful intermediate embeddings that do not map to any known token. Unless otherwise specified, **our results hold for a wide variety of models**, including Llama2-13B-Chat (Touvron et al., 2023), Llama3-8B (Dubey et al., 2024), Phi-3-Medium-4k-Instruct (Abdin et al., 2024), Gemma-1-7B (Gemma Team et al., 2024a), Gemma-2-9B (Gemma Team et al., 2024b), and Mistral-7B (Jiang et al., 2023).

## 4.1 SINGLE-TOKEN TIME CONTINUITY

We begin by studying how LLMs respond to variations in the time duration of a token. Consider the following input prompt to Llama3-8B:

```
In the sentence "apple  apple  apple  apple", how many fruits are mentioned?
                  d_s = 1  d_s = 1  d_s = 1  d_s = 1
```

The answer is naturally "4", and we expect language models to reply similarly.[2] However, suppose that we reduce the duration of the portion of the sentence between double quotes, as defined in Section 3), i.e.,

```
In the sentence "apple apple apple apple", how many fruits are mentioned?
                  ∑ d_s ∈ [1,4]
```

As the duration of the four "apple" tokens varies, we expect the model to output either (1) "4", since the number of tokens has not changed or (2) nonsensical/unrelated tokens, since such an input would be out-of-distribution. Instead, Llama 3 returns an output that both is meaningful and differs from that of the original sentence. As shown in Figure 2, the output consists of each number from 1 to 4, with peaks that vary with the token duration. In other words, the LLM interprets the sentence's content as if there were, respectively, 1, 2, 3 and 4 "apple" tokens.

From a linguistic perspective, the model returns outputs that, while reasonable, are hard to reconcile with the traditional interpretation of language in humans. LLMs naturally embody notions such as *half a token* (i.e., a token whose duration is not one time step), which humans lack.[3] This suggests that LLMs interpret language differently compared to humans.

We quantitatively study this phenomenon by repeating this experiment on a dataset of 200 word counting tasks. In particular, we reduce the duration of all repeated tokens (in our example, 'apple') and measure how many unique applicable peaks we observe as the duration varies. For instance, if the word is repeated four times, as we vary the duration factor we expect the model to count, for different duration factors, 1, 2, 3, and 4 elements.[4] A discrete interpretation of LLMs would predict that the model consistently predicts the same number (i.e. 4) regardless of the duration factor. Instead, we observe on average 190% more unique predictions. In other words, our results are incompatible with a discrete interpretation of the behaviour of LLMs. Refer to Appendix C.1 for a more in-depth overview of our methodology and results.

In the next section, we stress the generality of this observation by applying the notion of time duration to multiple tokens and entire sentences.

## 4.2 BEYOND SINGLE-TOKEN TIME CONTINUITY

A natural extension of token-level time continuity involves changing the duration of entire linguistic units.

We first study how LLMs behave when summing 2-digit numbers, where the duration of one of the addends is reduced. Consider the following input to Llama2-13B.[5]

---

[2]Indeed, all the models we studied replied with "4".

[3]To obtain the same output from humans, one would modify the input to include instructions such as `Consider each apple word as half an apple`. We do not need to prompt an LLM with such instructions.

[4]We treat 0 as an unexpected peak since the semantics of a zero-duration sentence are ill-defined.

[5]We use Llama2 instead of Llama3 as the latter treats both digits as a single token.

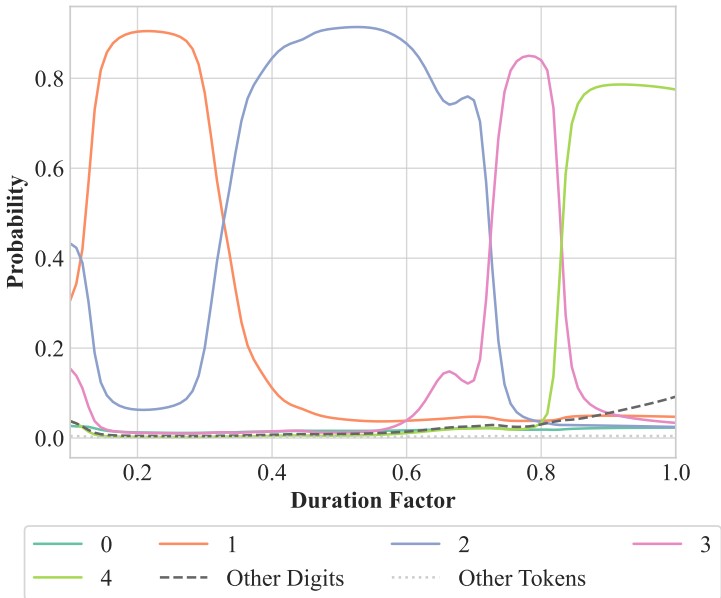

Figure 2: Time continuity experiments on the example in Section 4.1. If we reduce the time duration of the "apple" tokens, the transition between each output answer (a number between 1 and 4) is continuous.

```
The sum of 24 and   1   3   is
                    ⌣   ⌣
                  d_{s_1}  d_{s_2}
```

As shown in Figure 3a, by shrinking the duration of the tokens "1" and "3", we observe that the model transitions from predicting "3" (i.e., the first token of the sum $24 + 13$) to "2" (i.e., the model progressively treats "13" as a single-digit number). Refer to Appendix C.3 for more examples of this behaviour across models and choices of numbers.

Additionally, we study how LLMs behave when we reduce the duration of entire sentences. For instance, consider the following passage, which is fed to Llama3-8B:

```
Alice goes to the shop.

She buys a carton of milk.  She buys an apple.
⎵⎵⎵⎵⎵⎵⎵⎵⎵⎵⎵⎵⎵⎵⎵⎵⎵  ⎵⎵⎵⎵⎵⎵⎵⎵⎵⎵⎵⎵
         d_{s_1}                  d_{s_2}
She buys a potato.  She buys a loaf of bread.
⎵⎵⎵⎵⎵⎵⎵⎵⎵⎵⎵⎵⎵  ⎵⎵⎵⎵⎵⎵⎵⎵⎵⎵⎵⎵⎵⎵⎵
      d_{s_3}                d_{s_4}
How many items did Alice buy?
```

By reducing the duration of each sentence $\{d_{s_1}, d_{s_2}, d_{s_3}, d_{s_4}\}$, we once again observe the model replying as if Alice bought 1, 2, 3 or 4 items (Figure 3b), which is consistent with our findings in the previous section. In particular, we observe on average 205% more unique predictions than what would be expected from a discrete interpretation of LLMs. Refer to Appendix C.2 for further data on this behaviour, which we observed to be present across all models.

There are thus reasons to believe that the notion of time continuity is innate in pretrained LLMs. Such a notion might emerge as a natural consequence of their nature as smooth function approximations, though further research in this direction is needed.

## 4.3 SPACE CONTINUITY IN PRETRAINED LLMS

In addition to time continuity, we study the nature of space continuity in LLMs.

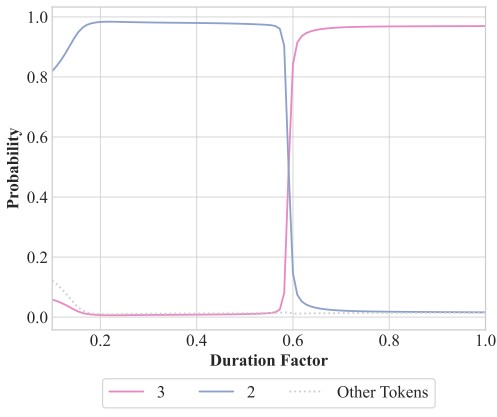
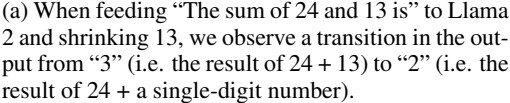

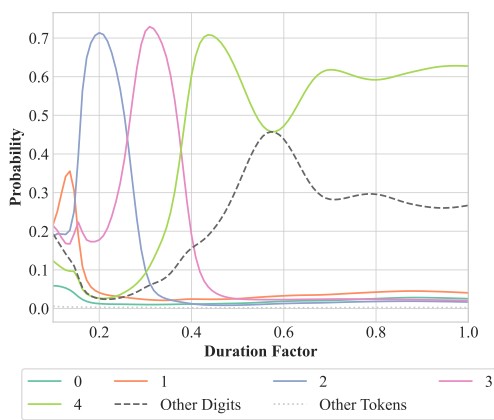

(a) When feeding "The sum of 24 and 13 is" to Llama 2 and shrinking 13, we observe a transition in the output from "3" (i.e. the result of 24 + 13) to "2" (i.e. the result of 24 + a single-digit number).

(b) If we reduce the time duration of each sentence, the transition between each output answer (a number between "1" and "4") is continuous.

Figure 3: Time continuity experiments as per Section 4.2 with Llama3-8B.

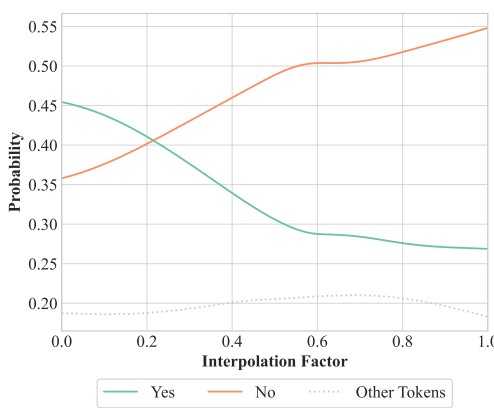

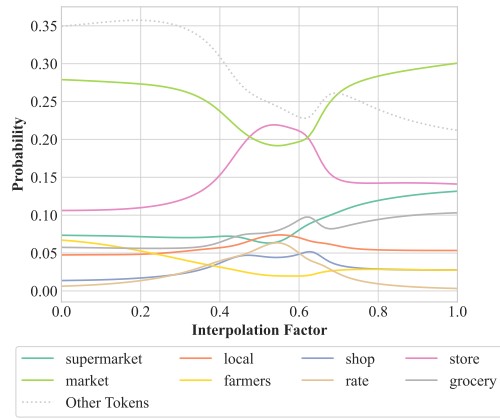

(a) Predicted next token for the sentence "Are ★ red?" for Llama 3.

(b) Predicted next token for the sentence "Alice bought some ★ at the" for Llama 3. All reported tokens have a probability of at least 5% at any point of the interpolation. For the other tokens, we report the cumulative distribution.

Figure 4: The linear embedding hypothesis holds for Llama3 and extends to the output prediction.

In the NLP literature, it is widely known that the embeddings of semantically similar tokens tend to share some of their semantics, an observation often referred as the *linear embedding hypothesis* (Mikolov et al., 2013b; Park et al., 2023). However, thanks to our generalisation we discover that such hypothesis also **holds for LLMs in the output space**. In other words, inputs undergo a series of non-linear transformations to make an LLM predict the next word; yet, the intermediate embeddings and the output preserve the semantics of the interpolated inputs, with the region of the embedding space that, while not having any meaningful interpretation, are treated as proper concepts with well-defined properties.

Consider the following example for Llama3-8B:

```
Are ★ red?
```

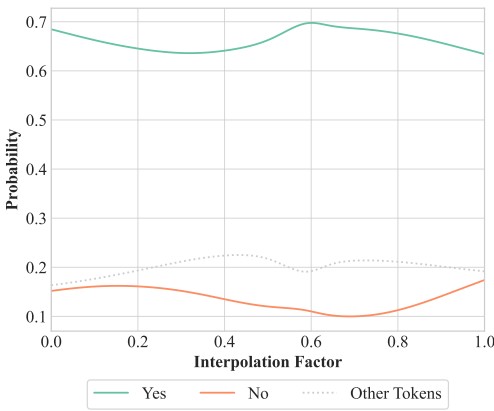

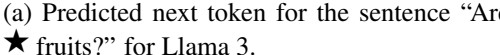

(a) Predicted next token for the sentence "Are ★ fruits?" for Llama 3.

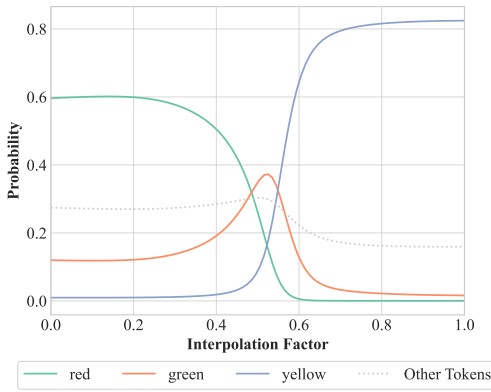

(b) Predicted next token for the sentence "The most common colour for ★ is". All reported tokens have a probability of at least 5% at any point of the interpolation. For the other tokens, we report the cumulative distribution.

Figure 5: Beyond the linear embedding hypothesis on Llama 3.

In the previous example, ★ represents a linear interpolation of the embeddings of the "apples" and "bananas" tokens. Intuitively, such an interpolation does not have a proper semantics in human language (there is no such thing as "apple-bananas"). In fact, the interpolation does not map to the embedding of any known token (see Appendix C.4). Nevertheless, Llama3's output (as well as other LLMs') smoothly transitions between outputting "yes" and "no", as shown in Figure 4 (left). This behaviour is further confirmed by similar prompts like "Alice bought some ★ at the" (Figure 4 (right)). In other words, any interpolation between "apples" and "bananas" is treated as a grammatically correct part of a sentence.

While one may argue that the context plays a role in modelling the range of possible answers to these questions (e.g., in the first example "yes" and "no" are the only reasonable options), this interpretation **only partially captures the nuances of LLMs' behaviour with space-continuous inputs**. In fact, if we prompt the model with the following input:

```
Are ★ fruits?
```

the model always replies "yes", as reported in Figure 5 (left). Any interpolation of "apples" and "bananas" is indeed a valid fruit for the LLM. Additionally, for a subset of models (namely Llama 3, Gemma 1, Mistral, and partially Gemma 2), when we feed the input:

```
The most common colour for ★ is
```

we discover that the intermediate colour of "apples" and "bananas" is "green" (Figure 5), i.e., these Transformers hold the knowledge that there is a fruit in between "apples" and "bananas" whose colour is neither "red" nor "yellow". Refer to Appendix C.4 for more quantitative and qualitative results.

In short, LLMs assign a semantic meaning to their embedding space, but this meaning is neither trivial nor consistent with human intuition. These results challenge the assumption that the way LLMs interpret language is a surrogate of that of humans.

## 5  IMPLICATIONS

Our results show that the current understanding of language models is incomplete. In this section, we highlight what we believe are promising implications of our findings.

**LLMs learn a continuous representation of language.** While LLMs are mostly trained on discrete data, they learn representations that go beyond the meaning and interplay of each token. We thus argue that our generalisation of Transformers opens up to new inference techniques: for example, multiple sub-tokens can be generated in parallel as interpolations of sentences with similar templates but different semantics. The same intuition can be applied to training. Yet, it requires solving a non-trivial issue: tokens fed simultaneously and in "superposition" (e.g., an interpolation of multiple, semantically similar words) generate an output where each corresponding token should be disentangled and reassigned to its original sentence.

**Language models treat language differently than humans.** Most of the examples reported in this paper contradict human intuition about language. Duration deeply affects a model's output in ways humans hardly grasp. Similarly, LLMs assign meaning to interpolations of embeddings even when such representations do not map to well-defined concepts. These behaviours represent a significant deviation from how humans reason about language, one that cannot be explained by a simple lack of data. We believe that the field of linguistics, alongside that of Machine Learning, should embrace the challenge of studying *the continuous language of LLMs*, synthesising classic tools from linguistics and deep learning. For instance, the notion of space continuity is deeply intertwined with how LLMs interpret language: we believe that our generalisation of Transformers can help us better understand the strong performance of LLMs on low-frequency inputs, as well as explain why they fail on edge cases that are semantically meaningful to humans.

Overall, our findings suggest that understanding more in-depth the continuous nature of LLMs can both improve their performance and align their representations with human reasoning.

## 6 CONCLUSION

In this paper, we introduce a generalisation of causal Transformers to study how pretrained LLMs respond to continuous inputs. We characterise the notions of time and space continuity, which we use to show that the language LLMs learn diverges from that of humans. In particular, LLMs assign complex semantic meanings to otherwise meaningless interpolations of embeddings and treat duration as a key property of their language. These results suggest that our traditional understanding of LLMs is incomplete.

We hope that, by studying the continuous behaviour of LLMs, future work will be able to shed further insight into the *language of Large Language Models*, with implications for both linguistics and LLM design.

## ACKNOWLEDGEMENTS

This work was supported by the EPSRC Centre for Doctoral Training in Autonomous Intelligent Machines and Systems n. EP/Y035070/1, in addition to Microsoft Ltd and the Collegio Superiore di Bologna. Emanuele La Malfa is supported by the Alan Turing Institute. We would like to thank Aleksandar Petrov and Constantin Venhoff for their insightful discussions.

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

## A   DETAILED DERIVATION OF SECTION 3

### A.1   A RECAP ON DISCRETE TRANSFORMER

In this section, we briefly recall how classical causal transformers work. Let $X \in \mathbb{R}^{T \times d}$ be the matrix whose rows are $\mathbf{x}_t^T$. Then, each head of a causal attention mechanism computes a matrix $Y \in \mathbb{R}^{T \times d}$, such that:

$$Y = \texttt{Attention}(Q, K, V) := \text{softmax}\left(\frac{QK^T}{\sqrt{d}} + M\right)V, \tag{10}$$

where $Q, K \in \mathbb{R}^{T \times d_k}$, $V \in \mathbb{R}^{T \times d}$ are the queries, the keys, and the value, respectively, while $M \in \mathbb{R}^{T \times T}$ is the causal attention mask, which is introduced to ensure that each row $\mathbf{y}_t^T$ of $Y$ only depends on observations $\mathbf{x}_s$ with $s \leq t$. The matrices $Q, K, V$ are defined as:

$$Q = XW^{(q)^T}, \qquad K = XW^{(k)^T}, \qquad V = XW^{(v)^T}, \tag{11}$$

where $W^{(q)}$, $W^{(k)}$, and $W^{(v)}$ are the attention matrices that are learned during training, while the causal attention mask $M$ is defined as:

$$M_{t,s} = \begin{cases} 0 & \text{if } t \leq s, \\ -\infty & \text{if } t > s. \end{cases} \tag{12}$$

The multi-head version of this causal attention mechanism typically used in modern architecture is obtained by computing $H$ independent versions $\{Y_1, \ldots, Y_H\}$ of Equation 10, and defining a learnable matrix $W^{(o)} \in \mathbb{R}^{dH \times d}$, which combines them to obtain a single transformed observation matrix:

$$Y = \texttt{cat}(Y_1, \ldots, Y_H)W^{(o)^T}. \tag{13}$$

A continuous version of a multi-head attention network can be simply obtained by considering Equation 10 on a single timestep $t \in [0, T]$. Indeed, it can be rewritten as:

$$\mathbf{y}_t = \sum_{s=1}^{T} \text{softmax}\left(\left[\frac{QK^T}{\sqrt{d}} + M\right]_{t,s}\right)\mathbf{v}_s, \tag{14}$$

where:

$$\text{softmax}\left(\left[\frac{QK^T}{\sqrt{d}} + M\right]_{t,s}\right) = \frac{\exp\left(\left[\frac{QK^T}{\sqrt{d}} + M\right]_{t,s}\right)}{\sum_{s'=1}^{T} \exp\left(\left[\frac{QK^T}{\sqrt{d}} + M\right]_{t,s'}\right)}. \tag{15}$$

Consequently,

$$\begin{aligned} \text{softmax}\left(\left[\frac{QK^T}{\sqrt{d}} + M\right]_{t,s}\right) &\propto \exp\left(\left[\frac{QK^T}{\sqrt{d}}\right]_{t,s}\right) \cdot \exp\left[M\right]_{t,s} \\ &= \begin{cases} \exp\left(\frac{\mathbf{q}_t^T \mathbf{k}_s}{\sqrt{d}}\right) & \text{if } s < t, \\ 0 & \text{otherwise}, \end{cases} \end{aligned} \tag{16}$$

Where $\mathbf{q}_t^T$ and $\mathbf{k}_s$ are the $t$-th and the $s$-th rows of $Q$ and $K^T$, respectively. Altogether, the above equations imply that:

$$\mathbf{y}_t = \sum_{s=1}^{t} \frac{1}{Z_t} \exp\left(\frac{\mathbf{q}_t^T \mathbf{k}_s}{\sqrt{d}}\right) \mathbf{v}_s, \tag{17}$$

where $Z_t := \sum_{s'=1}^{t} \exp\left(\frac{\mathbf{q}_t^T \mathbf{k}'_s}{\sqrt{d}}\right)$ is a normalisation constant. The above formula is then used in Section 3 to define CCT.

## A.2 DERIVATION OF EQUATION 8

We recall that:

$$\mathbf{q}(t) = W^{(q)}\mathbf{x}(t) = \sum_{k=1}^{T} W^{(q)}\mathbf{x}_k \mathbb{1}_{[a_k,b_k]}(t), \tag{18}$$

$$\mathbf{k}(t) = W^{(k)}\mathbf{x}(t) = \sum_{k=1}^{T} W^{(k)}\mathbf{x}_k \mathbb{1}_{[a_k,b_k]}(t), \tag{19}$$

$$\mathbf{v}(t) = W^{(v)}\mathbf{x}(t) = \sum_{k=1}^{T} W^{(v)}\mathbf{x}_k \mathbb{1}_{[a_k,b_k]}(t). \tag{20}$$

Consequently,

$$
\begin{aligned}
\mathbf{y}(t) &= \int_0^t \frac{1}{Z_t} \exp\left(\frac{\mathbf{q}(t)^T \mathbf{k}(s)}{\sqrt{d}}\right) \mathbf{v}(s) ds \\
&\stackrel{(20)}{=} \int_0^t \frac{1}{Z_t} \sum_{k=1}^{T} \exp\left(\frac{\mathbf{q}(t)^T \mathbf{k}(s)}{\sqrt{d}}\right) W^{(v)}\mathbf{x}_k \mathbb{1}_{[a_k,b_k]}(s) ds \\
&= \sum_{k=1}^{T} \int_{a_k}^{b_k} \frac{1}{Z_t} \exp\left(\frac{\mathbf{q}(t)^T \mathbf{k}(s)}{\sqrt{d}}\right) W^{(v)}\mathbf{x}_k ds \\
&\stackrel{(19)}{=} \sum_{k=1}^{T} \frac{1}{Z_t} W^{(v)}\mathbf{x}_k \int_{a_k}^{b_k} \exp\left(\frac{\mathbf{q}(t)^T \sum_{k'=1}^{T} W^{(k)}\mathbf{x}_{k'} \mathbb{1}_{[a_{k'},b_{k'}]}(s)}{\sqrt{d}}\right) ds \\
&= \sum_{k=1}^{T} \frac{1}{Z_t} W^{(v)}\mathbf{x}_k \int_{a_k}^{b_k} \exp\left(\frac{\mathbf{q}(t)^T W^{(k)}\mathbf{x}_k}{\sqrt{d}}\right) ds \\
&\stackrel{(18)}{=} \sum_{k=1}^{T} \frac{1}{Z_t} W^{(v)}\mathbf{x}_k \exp\left(\frac{\left(\sum_{t'=1}^{T} W^{(q)}\mathbf{x}_{t'} \mathbb{1}_{[a_{t'},b_{t'}]}(t)\right)^T W^{(k)}\mathbf{x}_k}{\sqrt{d}}\right) \int_{a_k}^{b_k} ds \\
&= \sum_{k=1}^{T} \frac{1}{Z_t} W^{(v)}\mathbf{x}_k \exp\left(\frac{\left(W^{(q)}\mathbf{x}_{\bar{t}}\right)^T W^{(k)}\mathbf{x}_k}{\sqrt{d}}\right) (b_k - a_k) \\
&= \sum_{k=1}^{T} \frac{1}{Z_t} \exp\left(\frac{\mathbf{q}_{\bar{t}}^T \mathbf{k}_k}{\sqrt{d}}\right) \mathbf{v}_k d_k,
\end{aligned}
\tag{21}
$$

which proves Equation 8.

## B EXPERIMENTAL SETUP

We now describe the shared aspects of our experiments.

## B.1 Implementing a CCT

CCTs can be implemented with little effort by starting with the implementation of a regular transformer and applying three modifications:

1. Modifying it so that it accepts arbitrary embeddings, rather than only tokens;
2. Modifying it so that positional indices can be floating points, instead of only integers;
3. Adding support for custom floating-point attention masks.

In our experiments, we used HuggingFace, which natively supports 1. and 3. and can be easily adapted to support 2.

Note that the last modification is necessary in order to support non-standard durations. In fact, the Euler discretisation of the integral in Equation (4) is equivalent to regular attention with carefully chosen attention coefficients.

**Proof** The discretisation of Equation (4) is, assuming that we have $n$ samples at positions $p_1, \ldots, p_n$, which represent the value of a piecewise constant function defined over the intervals $(0, p_1], (p_1, p_2], \ldots, (p_{n-1}, p_n]$:

$$\mathbf{y}(t) = \sum_{i \in 1, \ldots, n} \frac{1}{Z_t} (p_i - p_{i-1}) \exp\left(\frac{\mathbf{q}(i)^T \mathbf{k}(p_i)}{\sqrt{d}}\right) \mathbf{v}(p_i), \tag{22}$$

with $p_0 = 0$. In other words, if we are using multiplicative attention coefficients, the discretisation is equivalent to applying attention coefficients of the form $p_i - p_i - 1$. Intuitively, this means that the further apart two samples are, the higher the weight of the latter sample.

Note that for additive coefficients we can simply bring $p_i - p_{i-1}$ inside the exponential:

$$\mathbf{y}(t) = \sum_{i \in 1, \ldots, n} \frac{1}{Z_t} \exp\left(\log(p_i - p_{i-1}) + \frac{\mathbf{q}(i)^T \mathbf{k}(p_i)}{\sqrt{d}}\right) \mathbf{v}(p_i), \tag{23}$$

which is equivalent to an additive coefficient of $p_i - p_{i-1}$.

**In Practice** At the implementation level, a sequence of $n$ elements with durations $d_1, \ldots, d_n$ and embeddings $e_1, \ldots, e_n$ is fed to the extended transformer as follows:

- The embeddings $e_1, \ldots, e_n$ are fed directly, rather than feeding the sequence as tokens and mapping them to embeddings;
- The positional encodings are defined such that no "holes" are left in the piecewise constant function. In other words, the position $p_i$ is defined as $p_i = \sum_{j=1}^{i-1} d_j$;
- The durations are encoded using the formulae described in Equations (22) and (23).

## B.2 Experiment Hyperparameters

Since we study the logits, we do not use any typically generation-related hyperparameters (e.g. temperature and top-k). Aside from those described in Appendix B.1, we do not perform any other modification. Experiment-specific parameters are reported in the respective subsections of Appendix C.4.

## C Continuity - Full Results

### C.1 Single-Token Continuity

#### C.1.1 Qualitative Results

For single-token continuity, we shrink the subset of considered tokens with a coefficient in the range $[0.1, 1]$.

Since the LLMs do not necessarily return a numeric value, all of the queries were wrapped with a prompt to coax the model into doing so. The template for our prompts is thus:

```
Question:  In the sentence "[REPEATED WORDS]", how many times is
[CATEGORY] mentioned?  Reply with a single-digit number
Answer:
```

For Gemma 1, Llama 2 and Mistral we used a slight variation, since they did not return a numeric output:

```
Question:  How many [CATEGORY] are listed in the sentence
"[REPEATED WORDS]"?  Reply with a single-digit number
Answer:
```

We used variations of these two prompts throughout most of this paper. See the source code for further information. Alongside apples, we also tested the same prompt with the word "cat" (category: "animal") and "rose" (category: "flower"). See Figures 6 to 8 for full results.

### C.1.2 QUANTITATIVE RESULTS

We reduce the duration of all the steps (following the procedure described in Appendix B) and measure the time sensitivity, i.e. the number $k$ of unique applicable token peaks divided by the expected number of peaks $n$. For instance, if there are 4 steps, we expect to see peaks for 1, 2, 3, and 4. Our dataset is composed of 200 sentences with the same template as Appendix C.1.1 Note that, for some sentence+tokenizer combinations, a single word might be split into multiple tokens. As our analysis focuses on single-token words, we ignore such cases. Refer to Table 1 for the percentages of considered results for each model.

We define a unique relative peak as a situation where, for at least one duration factor, a certain class is the top prediction. For example, if by varying the duration factor the top class becomes '1', then '2', then '1' again, then '3', the unique relative peaks will be $\{1, 2, 3\}$. In our case, we only consider the probabilities of numerical tokens.

We normalise the number of relative peaks by the number of expected peaks (e.g., if a word is repeated four times and we observe three unique relative peaks, the normalised frequency is $3/4 = 0.75$). Note that, if the discrete interpretations of LLMs held true, we would only observe one unique relative peak (since the prediction would be constant as the duration factor varies). We name the hypothetical frequency in case such a hypothesis held true the *counterfactual* normalised frequency.

We report in Table 2 the counterfactual and observed normalised frequencies for the various models, as well as the average per-sample ratio between the two. Overall, the observed peak frequency is significantly higher than the counterfactual frequency; these results are thus incompatible with a discrete interpretation of LLMs.

For the sake of completeness, we report in Table 3 the results where we only count the *expected* peaks as valid (i.e. if an element is repeated four times, only the unique relative peaks '1', '2', '3' and '4' are considered valid).

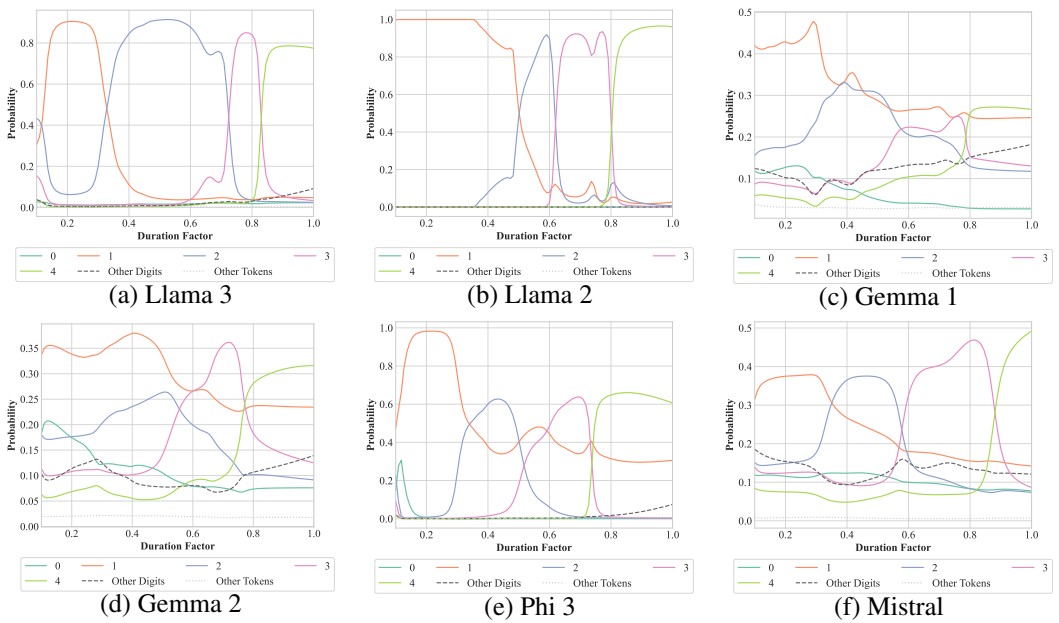

Figure 6: Predicted next token for the sentence "In the sentence 'apple apple apple apple', how many fruits are mentioned?" with duration shrinking for all studied models.

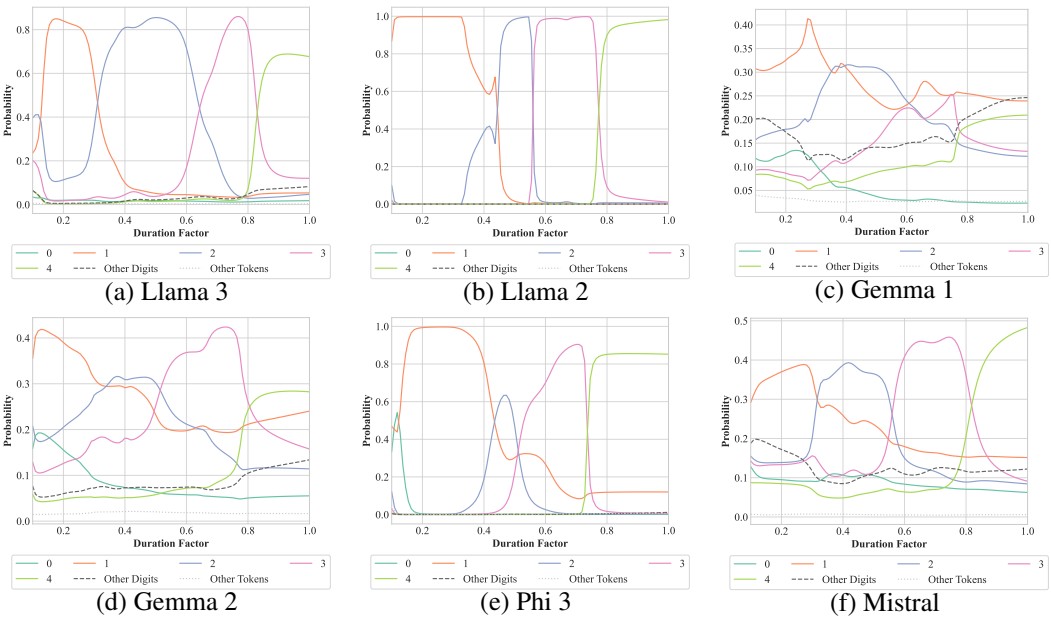

Figure 7: Predicted next token for the sentence "In the sentence 'cat cat cat cat', how many animals are mentioned?" with duration shrinking for all studied models.

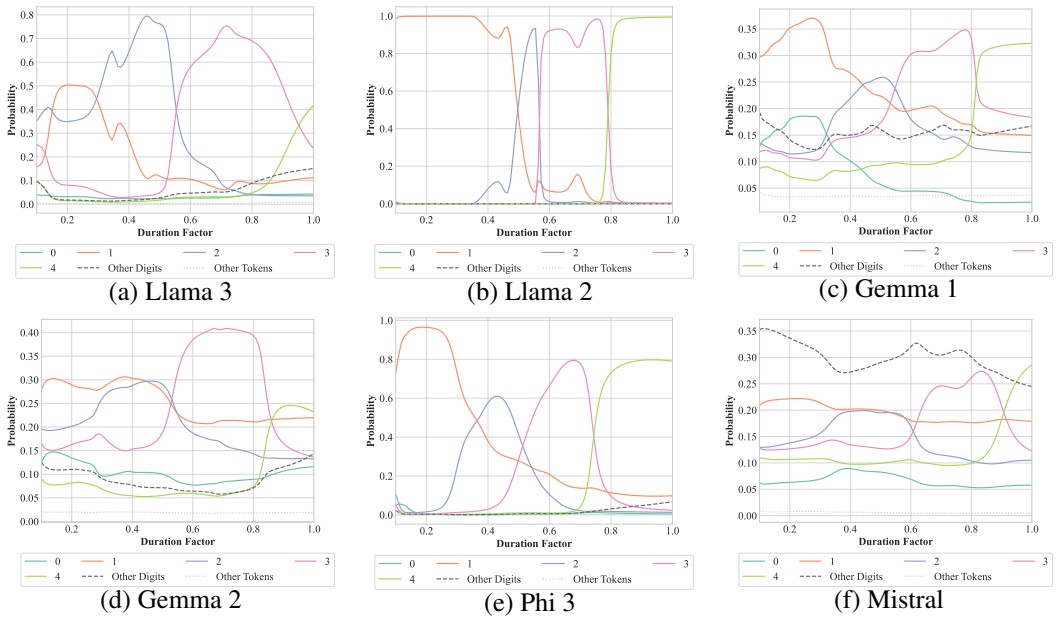

Figure 8: Predicted next token for the sentence "In the sentence 'rose rose rose rose', how many flowers are mentioned?" with duration shrinking for all studied models.

| Model Name | Valid Ratio |
|---|---|
| Llama 3 8b | 93.5% |
| Llama 2 13b | 54.5% |
| Gemma 1 7b | 100.0% |
| Gemma 2 9b | 100.0% |
| Phi 3 Medium | 54.5% |
| Mistral 7b | 76.0% |
| Global | 79.8% |

Table 1: Percentage of valid setups (i.e. setups where a word is treated as a single token) for the single token counting experiment. The 'Global' row refers to all valid records divided by all records.

| Model name | Normalised Peak Frequency | | Average Ratio |
|---|---|---|---|
| | Counterfactual | Observed | |
| Llama 3 8b | 0.2594 | 0.8953 | 3.57 |
| Llama 2 13b | 0.2599 | 0.9174 | 3.60 |
| Gemma 1 7b | 0.2610 | 0.5127 | 1.95 |
| Gemma 2 9b | 0.2610 | 0.8513 | 3.37 |
| Phi 3 Medium | 0.2599 | 0.8836 | 3.43 |
| Mistral 7b | 0.2584 | 0.4737 | 1.85 |
| Global | 0.2600 | 0.7404 | 2.90 |

Table 2: Counterfactual and observed normalised peak ratios for the single-token counting experiment, with all peaks (including unexpected ones). Note that the 'Global' row is an average weighted by the number of valid records.

| Model name | Normalised Peak Frequency | | Average Ratio |
|---|---|---|---|
| | Counterfactual | Observed | |
| Llama 3 8b | 0.2594 | 0.8914 | 3.56 |
| Llama 2 13b | 0.2599 | 0.8263 | 3.27 |
| Gemma 1 7b | 0.2610 | 0.3503 | 1.34 |
| Gemma 2 9b | 0.2610 | 0.8513 | 3.37 |
| Phi 3 Medium | 0.2599 | 0.792 | 3.09 |
| Mistral 7b | 0.2584 | 0.4737 | 1.85 |
| Global | 0.2600 | 0.6849 | 2.70 |

Table 3: Counterfactual and observed normalised peak ratios for the single-token counting experiment, with only expected peaks. Note that the 'Global' row is an average weighted by the number of valid records.

## C.2 COUNTING EVENTS

### C.2.1 QUALITATIVE RESULTS

Similarly to Appendix C.1, we used a prompt to coax the models into giving numeric outputs, as well as coefficients in the range $[0.1, 1]$. Alongside the shop example, we tested two other passages:

- The class went to the zoo. They saw a lion. They saw an elephant. They saw a giraffe. They saw a penguin. How many animals did the class see?
- Emily went to the beach. She found a seashell. She found a starfish. She found a smooth stone. She found a piece of seaweed. How many things did Emily find?

See Figures 9 to 11 for full results.

### C.2.2 QUANTITATIVE RESULTS

In addition to our qualitative results, we report further quantitative experiments for time duration.

We consider the sequential dataset from Lin et al. (2024), which contains 200 curated how-to tutorials split by step. Our template is as follows:

```
Tutorial: [Tutorial Title]
[Steps]
Question: How many steps are necessary to complete the tutorial?
Reply with a single-digit number
Answer: It takes
```

We then compute, in the same fashion as Appendix C.1.2, the normalised peak frequency, both when considering all peaks (Table 4) and only the expected peaks (Table 5).

| Model Name | Normalised Peak Frequency | | Average Ratio |
|---|---|---|---|
| | Counterfactual | Observed | |
| Llama 3 8b | 0.2186 | 0.7564 | 3.70 |
| Llama 2 13b | 0.2186 | 0.7274 | 3.42 |
| Gemma 7b | 0.2186 | 0.6967 | 3.46 |
| Gemma 2 9b | 0.2186 | 0.6188 | 3.21 |
| Phi 3 | 0.2186 | 0.7123 | 3.385 |
| Mistral | 0.2186 | 0.5119 | 2.52 |
| Global | 0.2186 | 0.6706 | 3.28 |

Table 4: Counterfactual and observed normalised peak ratios for the event counting experiment, with all peaks (including unexpected ones).

| Model Name | Normalised Peak Frequency | | Average Ratio |
|---|---|---|---|
| | Counterfactual | Observed | |
| Llama 3 8b | 0.2186 | 0.6397 | 3.26 |
| Llama 2 13b | 0.2186 | 0.4854 | 2.54 |
| Gemma 7b | 0.2186 | 0.5984 | 3.04 |
| Gemma 2 9b | 0.2186 | 0.5740 | 3.01 |
| Phi 3 | 0.2186 | 0.5436 | 2.75 |
| Mistral | 0.2186 | 0.4517 | 2.29 |
| Global | 0.2186 | 0.6097 | 3.05 |

Table 5: Counterfactual and observed normalised peak ratios for the event counting experiment, with only expected peaks.

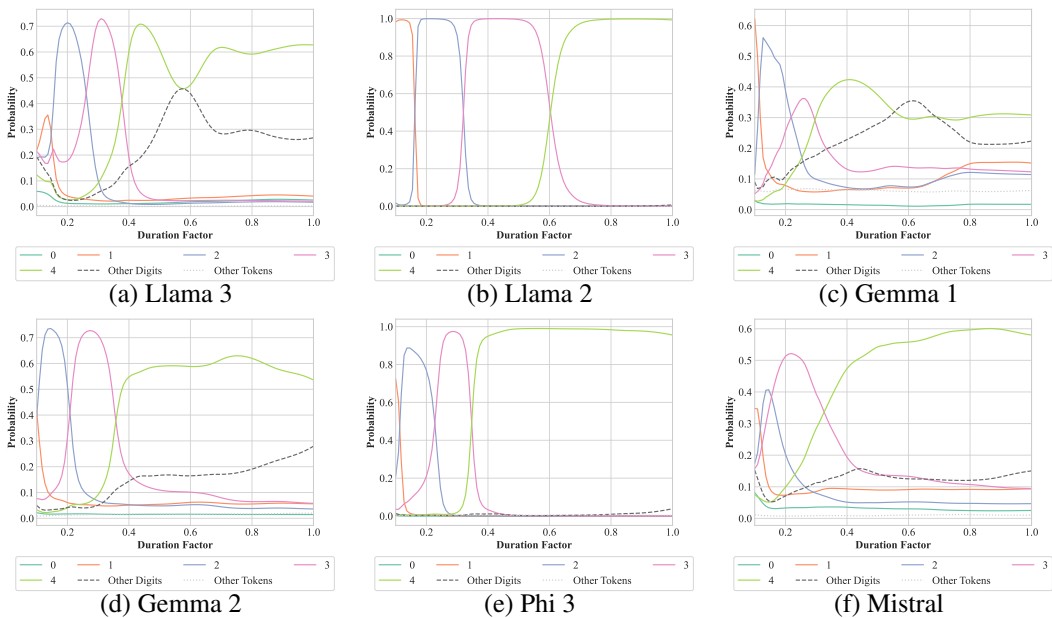

Figure 9: Predicted next token for the shop passage for all studied models, with duration shrinking.

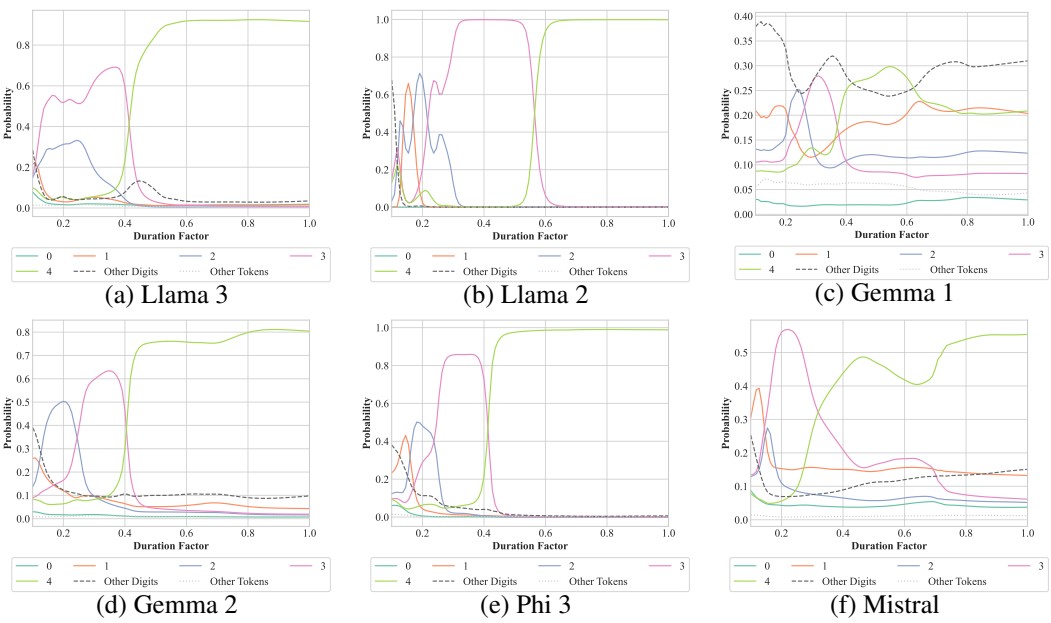

Figure 10: Predicted next token for the zoo passage for all studied models, with duration shrinking.

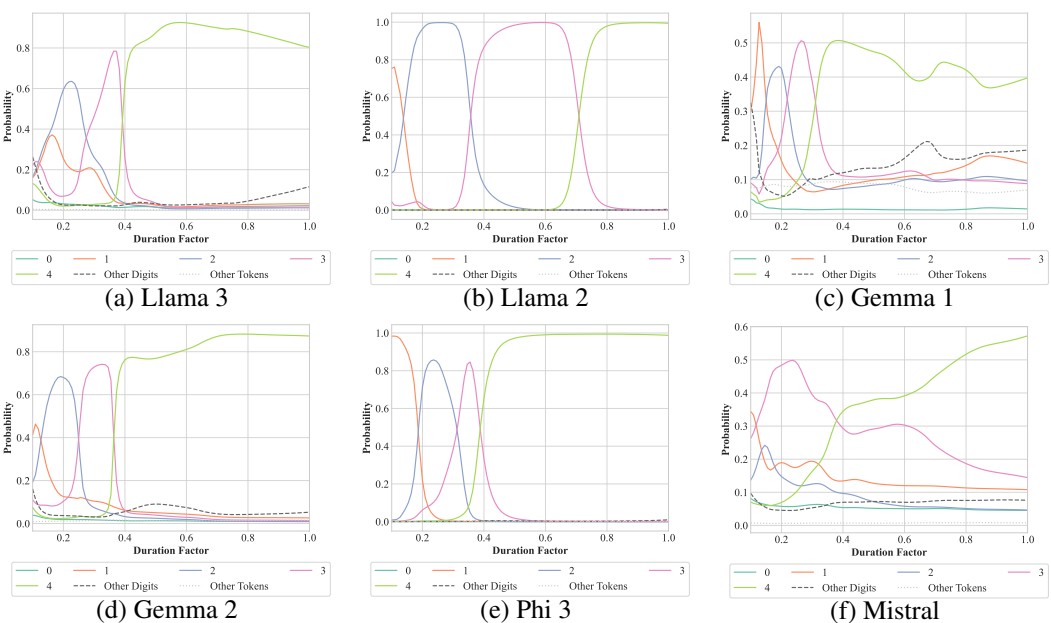

Figure 11: Predicted next token for the beach passage for all studied models, with duration shrinking.

| Model Name | Valid Rate |
|---|---|
| Llama 3 8b | 60.0% |
| Llama 2 13b | 19.5% |
| Gemma 1 7b | 94.5% |
| Gemma 2 9b | 95.0% |
| Phi 3 Medium | 48.5% |
| Mistral 7b | 93.0% |
| Global | 68.4% |

Table 6: Percentage of valid records (i.e. records where the unmodified output of the sum is correct) in the sums experiment.

### C.3 Number Sums

#### C.3.1 Qualitative Results

Our experimental setup is identical to that of Appendix C.1. In addition to 24 + 13, we repeat our experiments with the sums 13 + 74 and 32 + 56. Refer to Figures 12 to 14 for full results.

#### C.3.2 Quantitative Results

We consider a dataset of 100 questions involving sums, such as the following:

```
Question: Mia delivered 82 packages in the morning and 38 packages
in the afternoon. How many packages did Mia deliver in total?
Answer: Mia delivered
```

where `item_1` and `item_2` are two-digit numbers. The specific items and questions vary, but in all cases the answer is the sum of two two-digit numbers.

For each sentence, we independently shrink each of the two numbers, for a total of 200 records. In each record, we observe how the predicted probabilities vary as the duration factor varies. We only consider records where the model correctly computes the first digit of the sum on the unaltered sentence (see Table 6 for a per-model breakdown).

Let $y_o$ be the original label (i.e. the first digit of the sum of the two numbers) and $Y_s$ the *shrunk* labels (i.e. the set of predicted digits in case the shrunk number is treated as a single-digit number. For instance, in the sum 24 + 37, the original label is 6 (24 + 37 = 61), but the shrunk labels are 2 (24 + 3 = 27) and 3 (24 + 7 = 31). Note that we ignore non-numerical labels.

We then check three properties:

- **P1**: For a certain duration factor, the collective probability of the labels in $Y_s$ is higher than that of $y_o$;
- **P2**: For a certain duration factor, the collective probability of the labels in $Y_s$ is higher than that of $y_o$ and any other numerical label;
- **P3**: For a certain duration factor, the collective probability of the labels in $Y_s$ is higher than that of $y_o$ and any other numerical label. Additionally, at no point is another numerical label (i.e. neither $y_o$ nor an element of $Y_s$) the top label.

Note that P3 implies P2 and that P2 implies P1.

We report how frequently each property is true in Table 7.

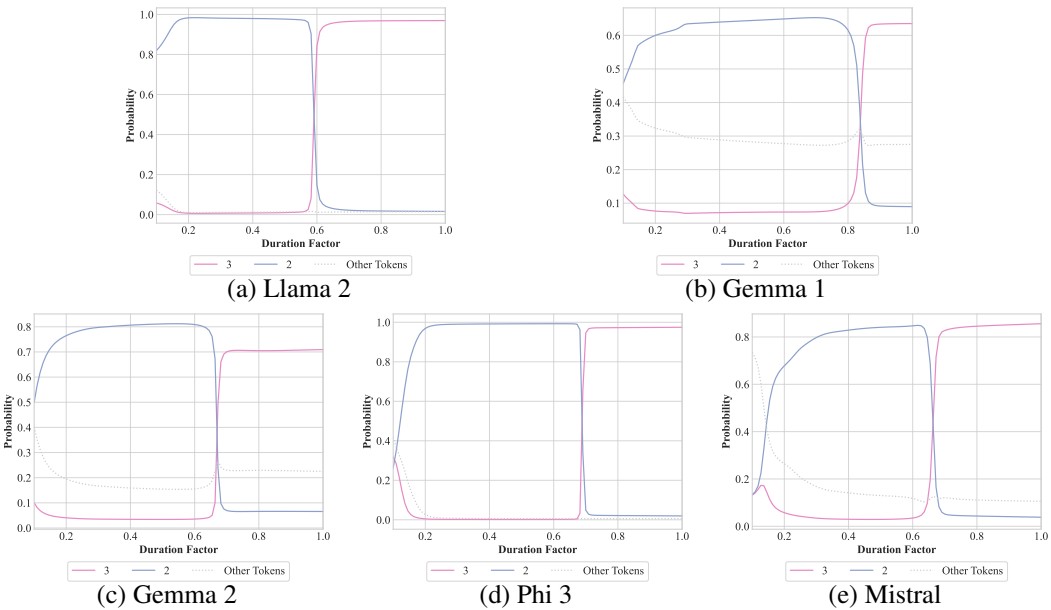

Figure 12: Predicted next token for the sentence "The sum of 24 and 13 is" in all studied models (except Llama 3) with shrinking of 13.

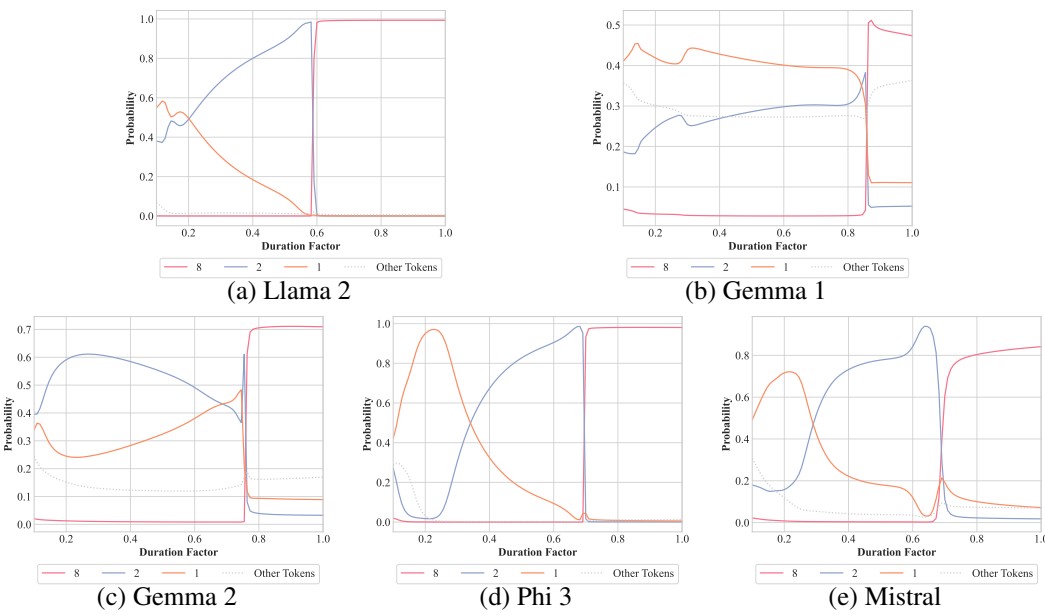

Figure 13: Predicted next token for the sentence "The sum of 13 and 74 is" in all studied models (except Llama 3) with shrinking of 74.

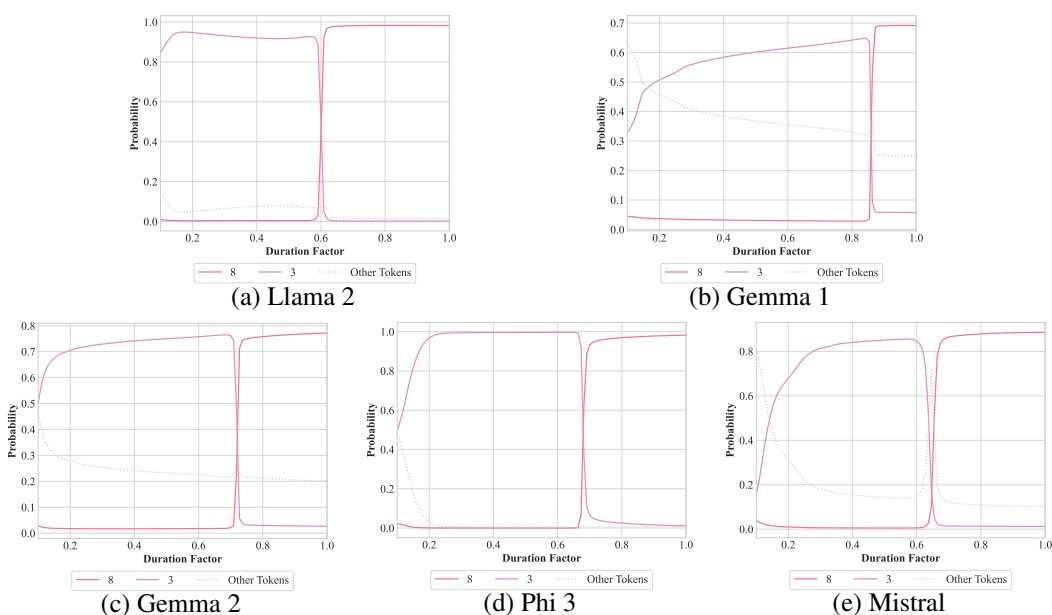

Figure 14: Predicted next token for the sentence "The sum of 32 and 56 is" in all studied models (except Llama 3) with shrinking of 32.

| Model Name | P1 Frequency | P2 Frequency | P3 Frequency |
|---|---|---|---|
| Llama 3 8b | 29.17% | 25.83% | 25.83% |
| Llama 2 13b | 92.31% | 87.18% | 76.92% |
| Gemma 1 7b | 97.88% | 97.35% | 90.48% |
| Gemma 2 9b | 95.79% | 95.79% | 89.47% |
| Phi 3 Medium | 100.00% | 100.00% | 87.63% |
| Mistral 7b | 100.00% | 100.00% | 87.10% |
| Global | 87.82% | 86.97% | 79.05% |

Table 7: Frequency of each property for valid records in the sums experiment. Note that the 'Global' row is an average that is weighted by the number of valid records.

### C.4    SPACE CONTINUITY

#### C.4.1    QUALITATIVE RESULTS

We report the full results concerning interpolations of embeddings in the main paper in Figures 15 to 17 and 19. We also check that the intermediate interpolation does not correspond to any existing token by asking the LLM to repeat the embedding:

`Repeat the word ★.`

As shown in Figure 19, the repetition of ★ does not correspond to any existing token (as shown by the lack of peaks for tokens other than those related to apples and bananas).

Additionally, we adapt some experiments to another pair of tokens, namely cats and dogs, where we find that the interpolation of cats and dogs is an animal, but whether cats-dogs meow depends on the position along the interpolation axis (see Figures 20 to 23). Similarly, refer to Figures 24 to 27 for our results on the water-juice interpolation.

#### C.4.2    BOOLEAN INTERPOLATION

We then test how our results compare with studies on interpolation of Boolean formulae. To do so, we perform linear interpolations of Boolean binary operators and study how intermediate operators behave. In particular, we study interpolations of:

- AND and OR;
- AND and XOR;
- AND and NAND;
- OR and NOR.

We report our results for all models whose tokenizers treat the operators as having the same number of tokens. While the models often struggle to compute the correct Boolean results for discrete inputs, we nonetheless observe the emergence of "fuzzy" operators, whose truth values can be best represented as floating points.

#### C.4.3    QUANTITATIVE RESULTS

We consider 50 pairs of objects having some properties in common (e.g. apples and bananas are both fruits). For each pair, we consider one common property, one property shared by one element of the pair, one property shared by the other element of the pair, and one property that is shared by neither of them, for a total of 200 records. We then interpolate (with 40 steps) between the sentence containing one object or the other. For instance, for a property shared by both apples and bananas, we interpolate between the sentences:

`Question: Can apples be eaten raw? (yes/no)\nAnswer:`
`Question: Can bananas be eaten raw? (yes/no)\nAnswer:`

and compute the predicted scores of 'yes' and 'no'.

Note that some questions may have borderline answers (i.e. the answer could be argued to be true or false); we still consider such questions valid, as we are interested in the variation of the output throughout the interpolation rather than the specific answer. We however ignore object-tokenizer pairs where the two resulting sentences have different tokenized lengths (as that prevents interpolation). We report the percentage of valid records in Table 8.

To measure the smoothness of the variation in output, we compute the maximum of the absolute derivative across the interpolation interval, i.e. $\max_{x \in [a,b]} |f'(x)|$ (which can be seen as an estimate of the Lipschitz constant). We then normalize by dividing by the amplitude (i.e. $\max_{x \in [a,b]} f(x) - \min_{x \in [a,b]} f(x)$). Although imperfect, this metric provides insight into the 'sharpest' variation in output for a model. We report the average metric in Table 9. In general, we observe that Llama 2 and Gemma 2 have higher normalised maximum absolute derivatives, while the other models tend to have similar normalised values.

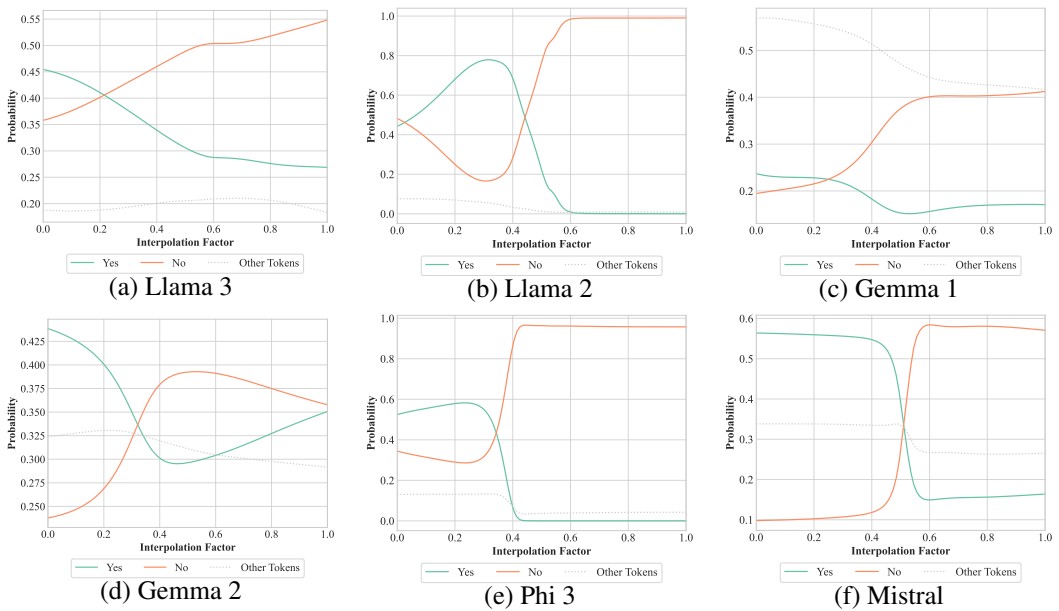

Figure 15: Predicted next token for the sentence "Are ★ red?", where ★ is an interpolation of "apples" and "bananas".

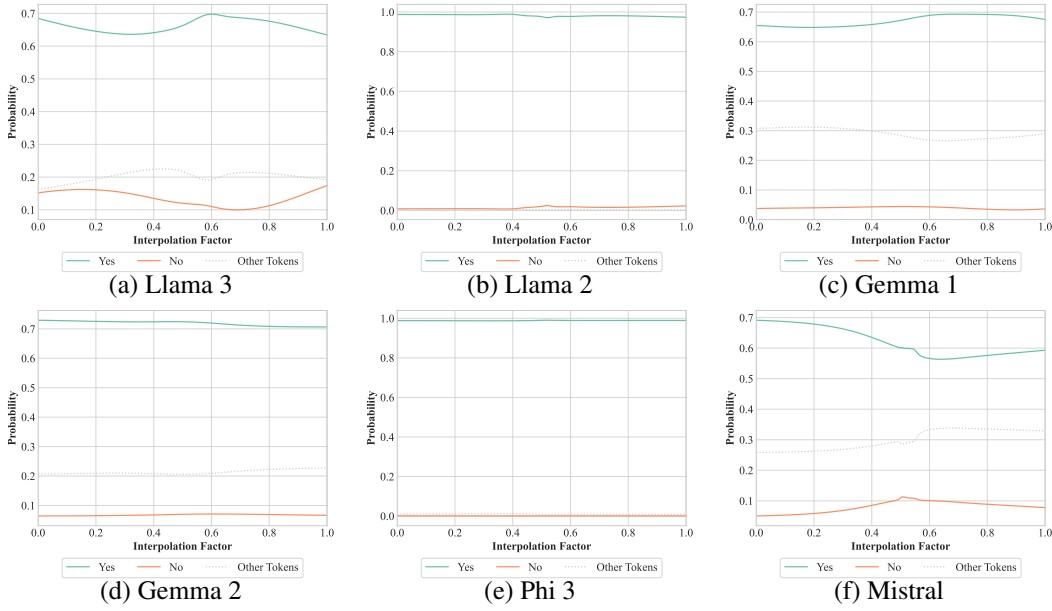

Figure 16: Predicted next token for the sentence "Are ★ a fruit?", where ★ is an interpolation of "apples" and "bananas".

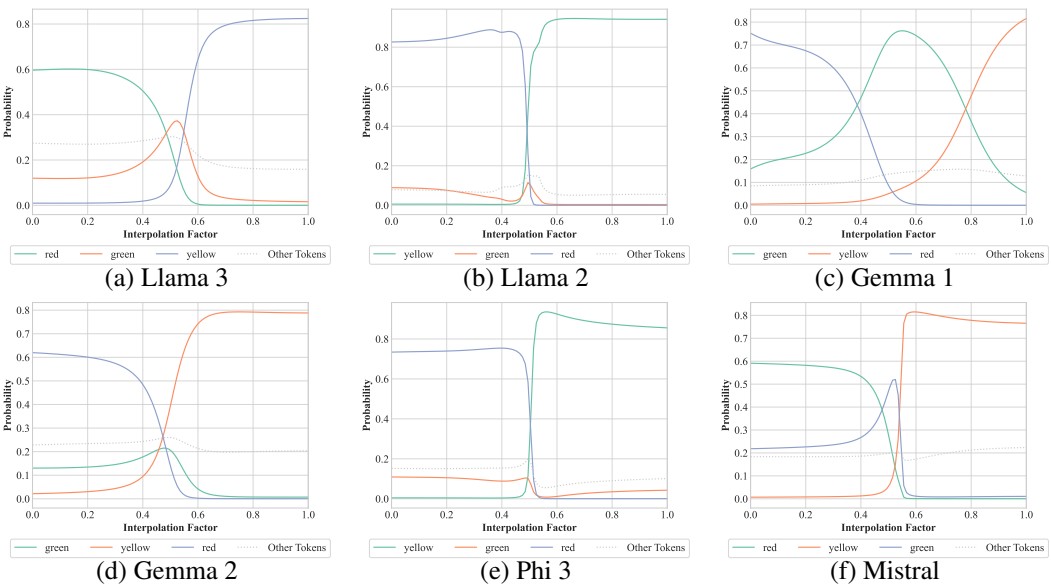

Figure 17: Predicted next token for the sentence "The most common colour of ★ is", where ★ is an interpolation of "apples" and "bananas". We report all tokens with a probability of at least 5% at any point of the interpolation.

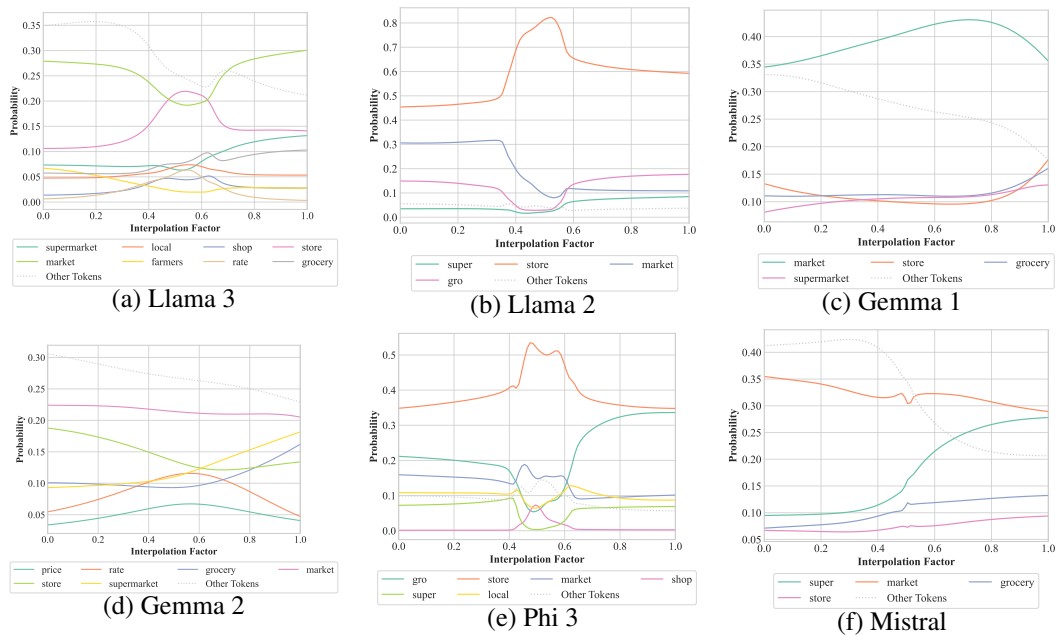

Figure 18: Predicted next token for the sentence "Alice bought some ★ at the", where ★ is an interpolation of "apples" and "bananas". We report all tokens with a probability of at least 5% at any point of the interpolation.

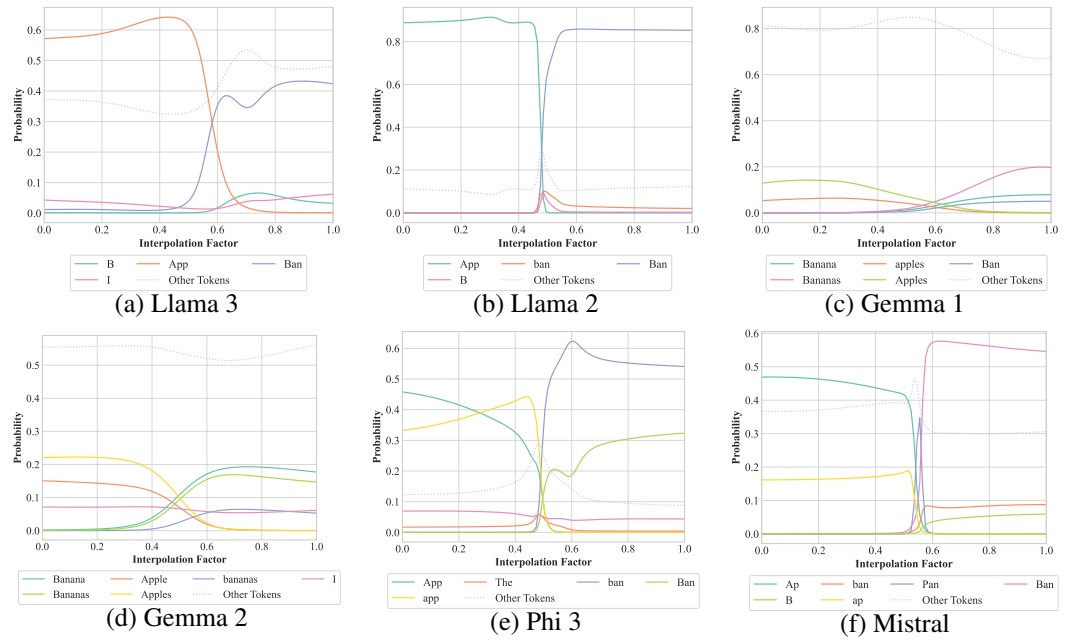

Figure 19: Predicted next token for the sentence "Repeat the word ★", where ★ is an interpolation of "apples" and "bananas". We report all tokens with a probability of at least 5% at any point of the interpolation.

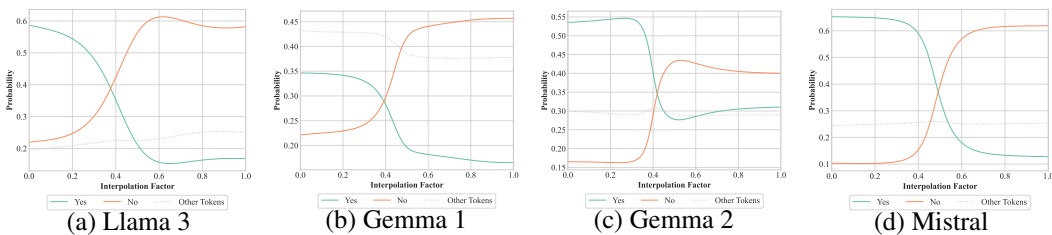

Figure 20: Predicted next token for the sentence "Do ★ meow?", where ★ is an interpolation of "cats" and "dogs". Results for Llama 2 and Phi 3 are not reported due to the two sentences having a different number of tokens.

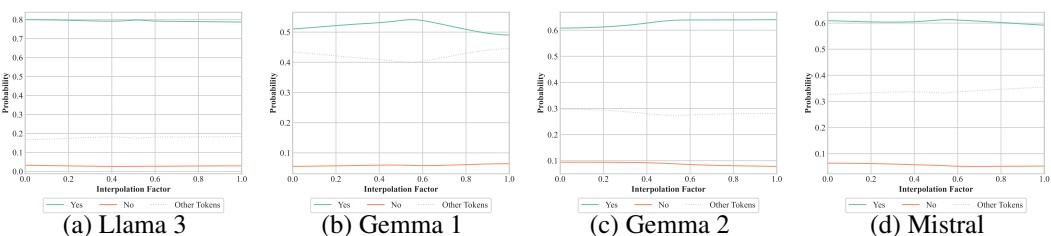

Figure 21: Predicted next token for the sentence "Are ★ animals?", where ★ is an interpolation of "cats" and "dogs". Results for Llama 2 and Phi 3 are not reported due to the two sentences having a different number of tokens.

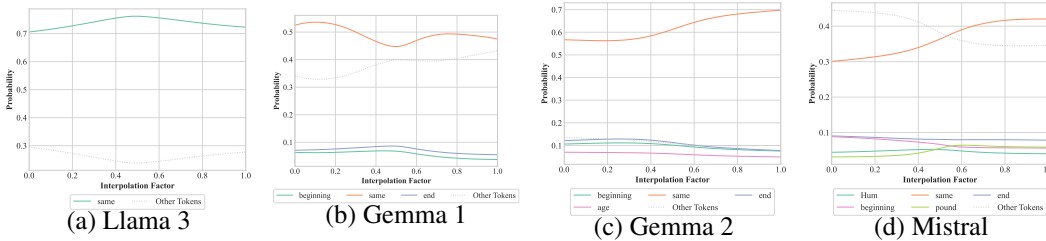

Figure 22: Predicted next token for the sentence "We bought two ★ at the", where ★ is an interpolation of "cats" and "dogs". We report all tokens with a probability of at least 5% at any point of the interpolation. Results for Llama 2 and Phi 3 are not reported due to the two sentences having a different number of tokens.

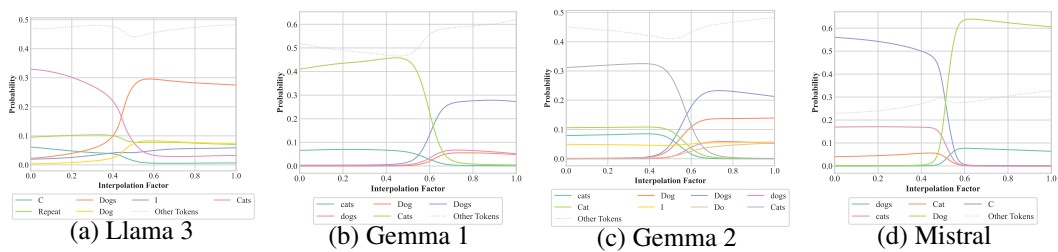

Figure 23: Predicted next token for the sentence "Repeat the word ★.", where ★ is an interpolation of "cats" and "dogs". We report all tokens with a probability of at least 5% at any point of the interpolation. Results for Llama 2 and Phi 3 are not reported due to the two sentences having a different number of tokens.

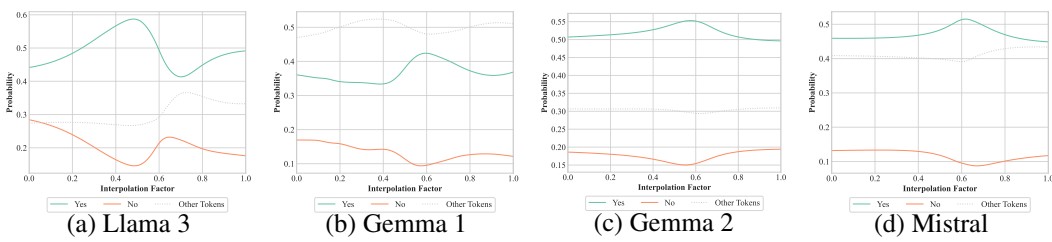

Figure 24: Predicted next token for the sentence "Does ★ contain sugar?", where ★ is an interpolation of "water" and "juice". Results for Llama 2 and Phi 3 are not reported due to the two sentences having a different number of tokens.

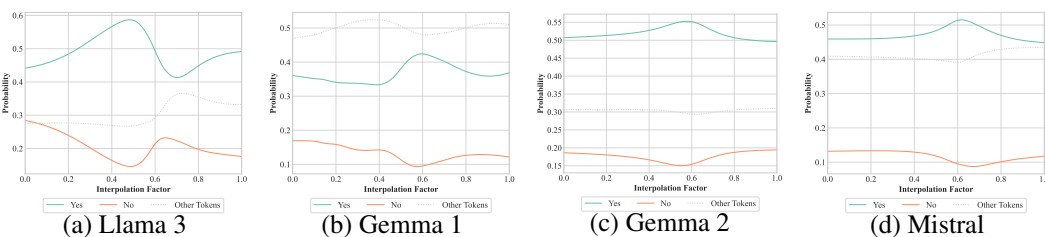

Figure 25: Predicted next token for the sentence "Is ★ a drink?", where ★ is an interpolation of "water" and "juice". Results for Llama 2 and Phi 3 are not reported due to the two sentences having a different number of tokens.

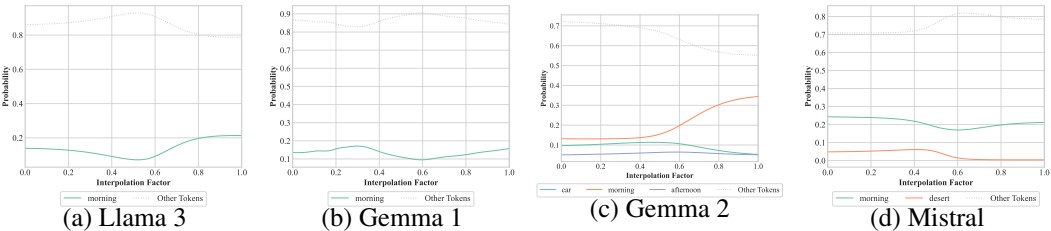

Figure 26: Predicted next token for the sentence "We drank some ★ in the", where ★ is an interpolation of "water" and "juice". We report all tokens with a probability of at least 5% at any point of the interpolation. Results for Llama 2 and Phi 3 are not reported due to the two sentences having a different number of tokens.

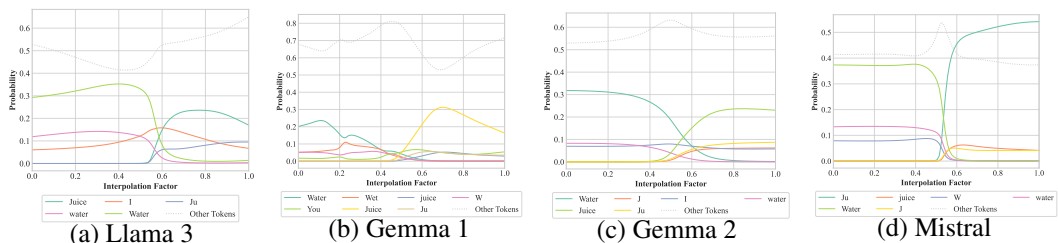

Figure 27: Predicted next token for the sentence "Repeat the word ★.", where ★ is an interpolation of "water" and "juice". We report all tokens with a probability of at least 5% at any point of the interpolation. Results for Llama 2 and Phi 3 are not reported due to the two sentences having a different number of tokens.

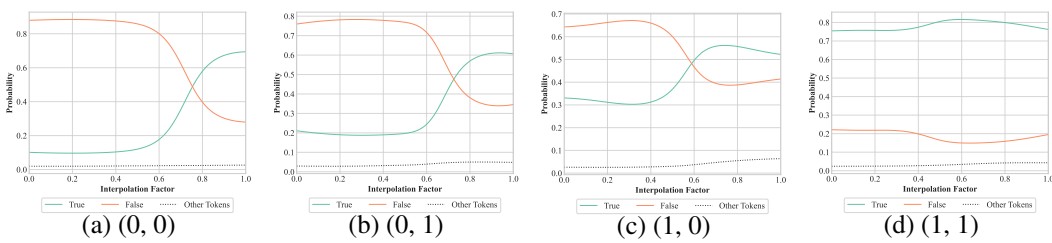

Figure 28: Predicted next token for interpolations of AND and OR in Llama 3.

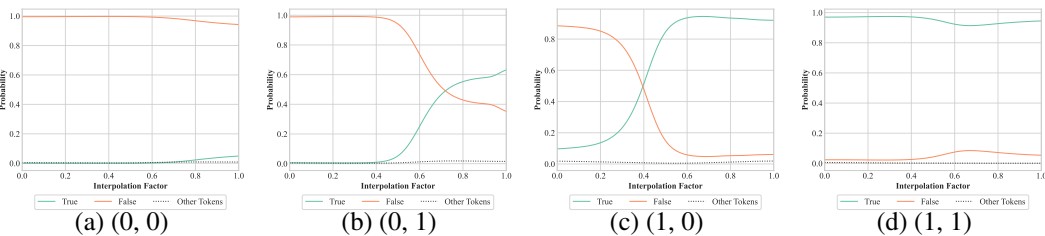

Figure 29: Predicted next token for interpolations of AND and OR in Llama 2.

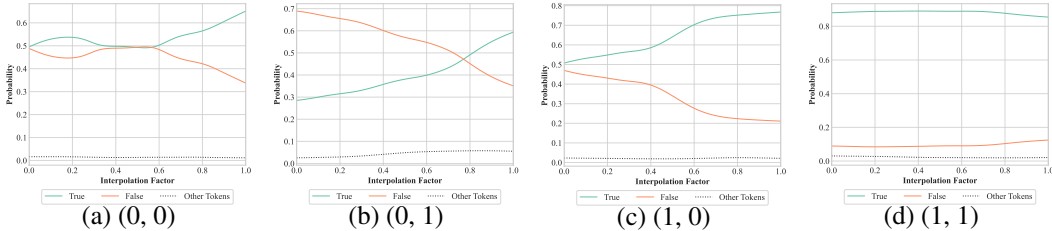

Figure 30: Predicted next token for interpolations of AND and OR in Gemma.

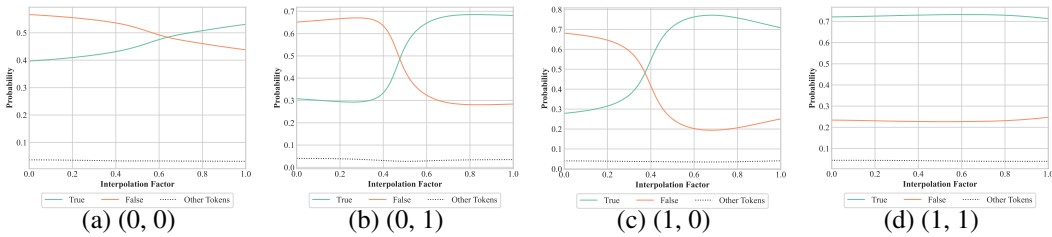

Figure 31: Predicted next token for interpolations of AND and OR in Gemma 2.

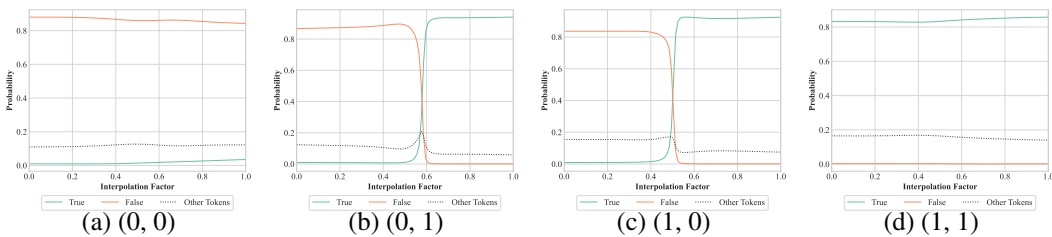

Figure 32: Predicted next token for interpolations of AND and OR in Phi 3.

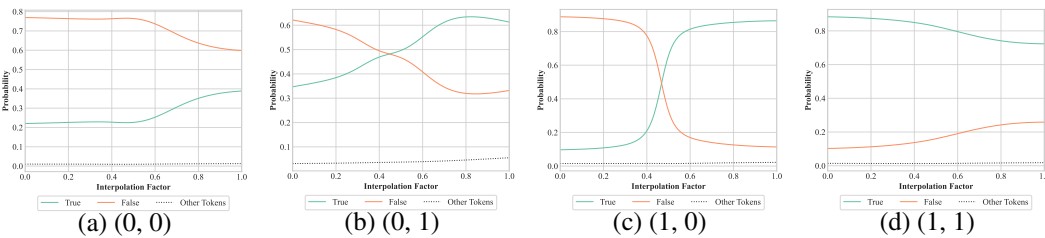

Figure 33: Predicted next token for interpolations of AND and OR in Mistral.

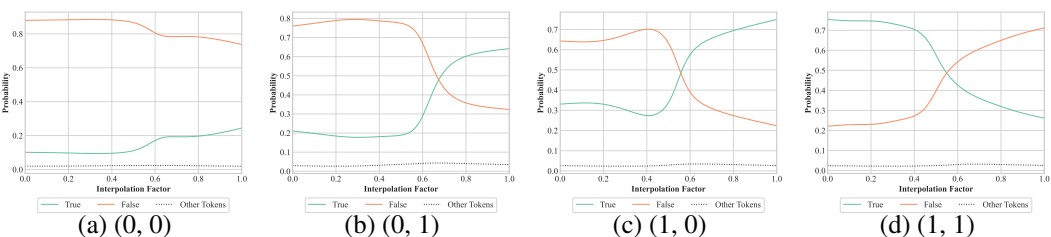

Figure 34: Predicted next token for interpolations of AND and XOR in Llama 3.

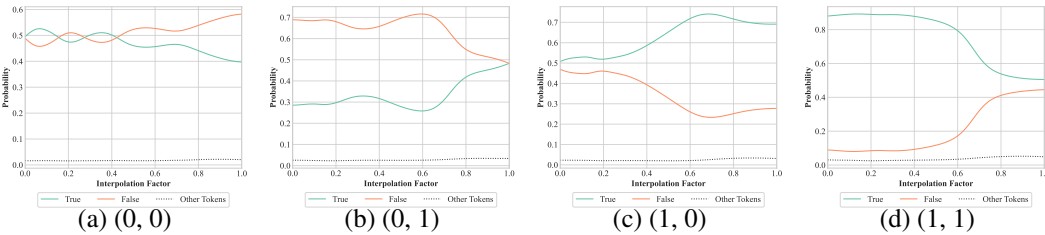

Figure 35: Predicted next token for interpolations of AND and XOR in Gemma.

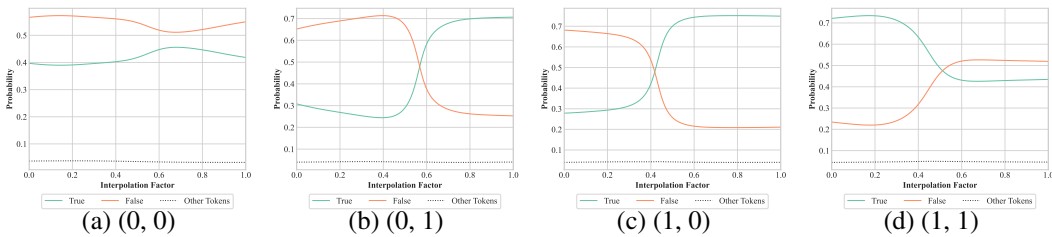

Figure 36: Predicted next token for interpolations of AND and XOR in Gemma 2.

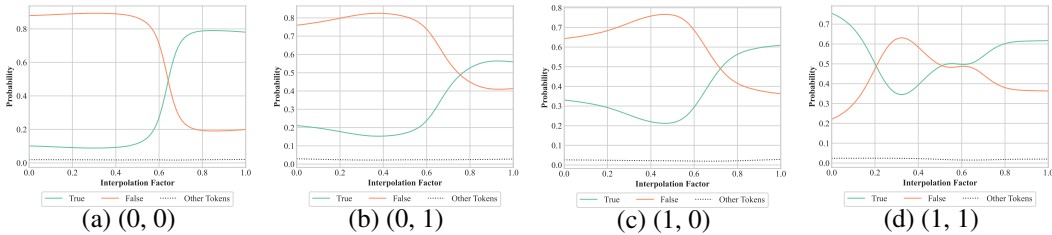

Figure 37: Predicted next token for interpolations of AND and NAND in Llama 3.

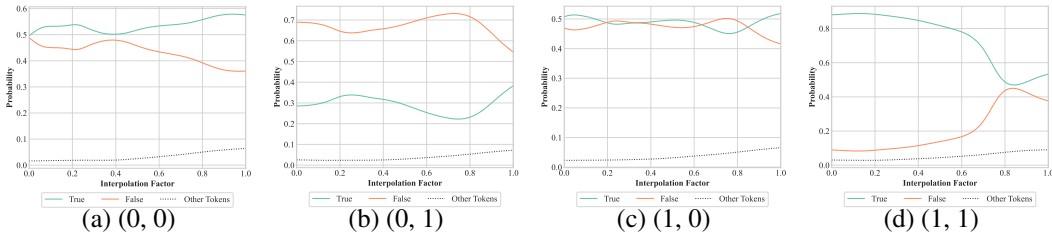

Figure 38: Predicted next token for interpolations of AND and NAND in Gemma.

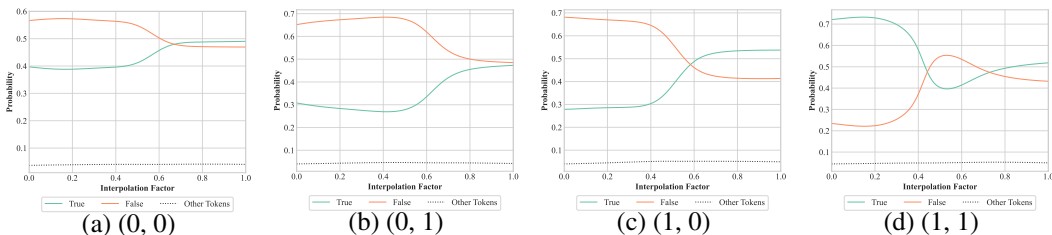

Figure 39: Predicted next token for interpolations of AND and NAND in Gemma 2.

| Model Name | Valid Rate |
|---|---|
| Llama 3 8b | 74.0% |
| Llama 2 13b | 34.0% |
| Gemma 1 7b | 84.0% |
| Gemma 2 9b | 84.0% |
| Phi 3 Medium | 34.0% |
| Mistral 7b | 46.0% |
| Global | 59.3% |

Table 8: Percentage of valid records (i.e. where both sentences have the same tokenized length) in the embedding interpolation experiment.

| Model Name | Normalised Maximum Absolute Derivative |
|---|---|
| Llama 3 8b | 4.959 |
| Llama 2 13b | 9.005 |
| Gemma 1 7b | 5.090 |
| Gemma 2 9b | 5.402 |
| Phi 3 Medium | 9.279 |
| Mistral 7b | 7.445 |
| Global | 6.214 |

Table 9: Average normalised maximum absolute derivative for each model. The "Global" row is an average weighted by the number of valid records.

We also study whether the intermediate embeddings have outputs that are significantly different from those of the embedding extremes. In theory, we would expect the probability of an output computed on an intermediate embedding to be an interpolation between the value for one object or the other. Let $f(a)$ be the probability score for one extreme of the interpolation and $f(b)$ the probability score for the other extreme. We consider our expected range as $[\min(f(a), f(b)), \max(f(a), f(b))]$. We then verify whether, for some intermediate value, the predicted probability of 'yes' or 'no' is outside their respective ranges.[6] In particular, we compute the maximum difference between the allowed range and the actual value, i.e.:

$$m_{\text{diff}} = \max_{x \in [a,b]} \begin{cases} |f(x) - \min(f(a), f(b))| & f(x) < \min(f(a), f(b)) \\ |f(x) + \max(f(a), f(b))| & f(x) > \max(f(a), f(b)) \end{cases} \tag{24}$$

We define $m_{\max}$ as the maximum of the $m_{\text{diff}}$ values for 'yes' and 'no'. Note that if the function only has values in $[\min(f(a), f(b)), \max(f(a), f(b))]$, $m_{\max}$ will be 0. We report the average $m_{\max}$ and the percentage of records where $m_{\max} \geq 0.05$ in Table 10. We also report the histograms describing the distributions of $m_{\max}$ in Figure 40. In general, we observe on average that 20% of records have an $m_{\max}$ higher than 0.05, which cannot be explained by a simple interpolation-based model of LLM behaviour.

## D   ADDITIONAL RESULTS

We complement our previous observations on time continuity with experiments on two common sequence transformations, namely shifting and scaling.

---

[6]Note that increasing the probability of 'yes' does not necessarily decrease the probability of 'no', as the model can also provide spurious outputs.

| Model Name | Average $m_{\mathbf{max}}$ | % ($m_{\mathbf{max}} \geq 0.05$) |
|---|---|---|
| Llama 3 8b | 0.0434 | 32.43% |
| Llama 2 13b | 0.0639 | 23.53% |
| Gemma 1 7b | 0.0332 | 22.02% |
| Gemma 2 9b | 0.0178 | 9.52% |
| Phi 3 Medium | 0.0212 | 14.71% |
| Mistral 7b | 0.0362 | 22.83% |
| Global | 0.0339 | 20.79% |

Table 10: Average $m_{\max}$ for each model and percentage of records where $m_{\max} > 0.05$. The "global" row is an average weighted by the number of valid records.

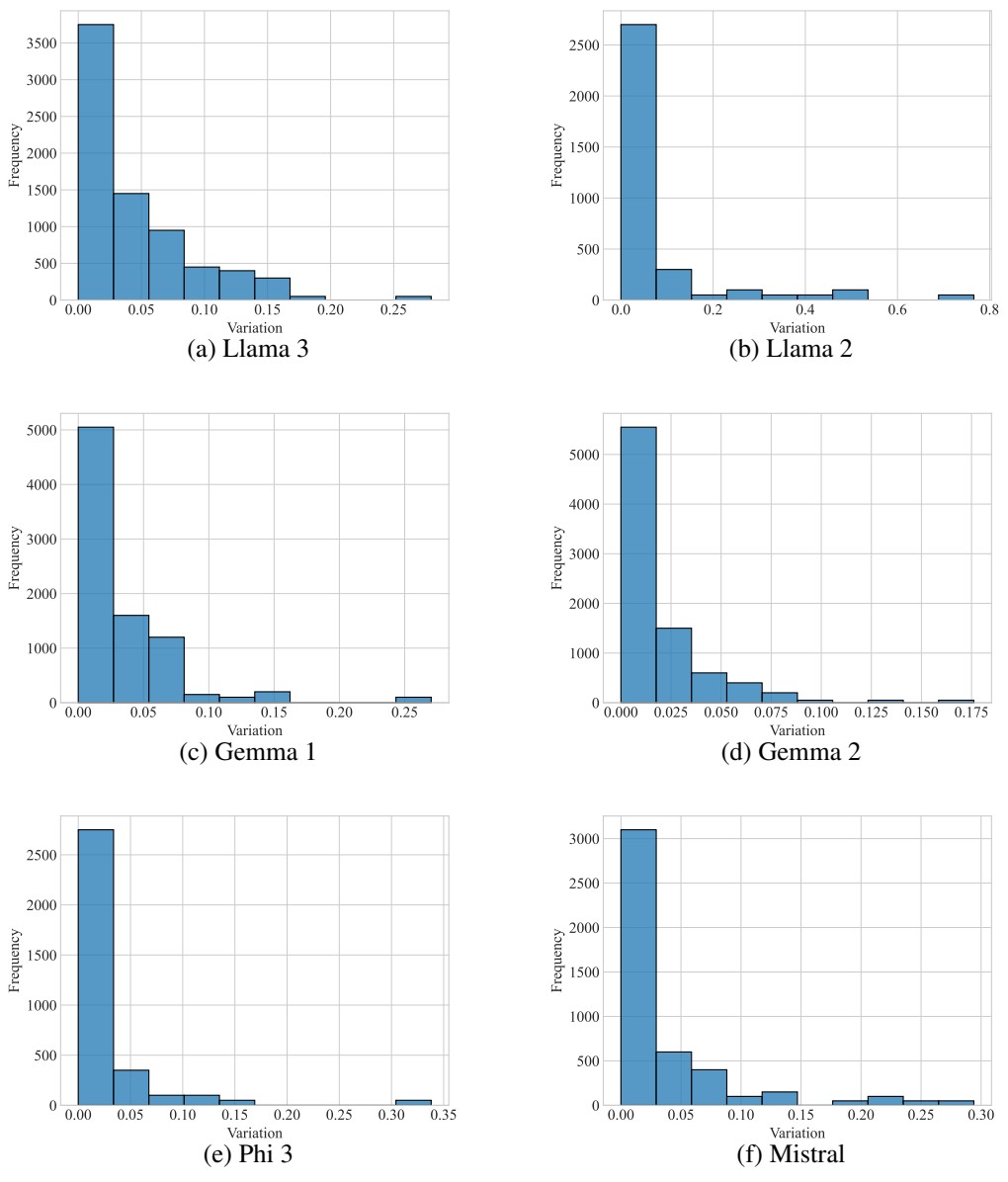

Figure 40: Distribution of $m_{\max}$ for all the studied models.

## D.1 TRANSLATIONAL INVARIANCE

For shifting, we increment the positional embeddings of each token (as well as the lower bound of integration) by a fixed amount (up to 10) without actually changing the sentence's duration. In particular, we feed the following input to Llama3-8B:

$$\underbrace{\text{The}}_{\xrightarrow{\Delta d} d_{s_1}} \quad \underbrace{\text{capital}}_{\xrightarrow{\Delta d} d_{s_2}} \quad \underbrace{\text{of}}_{\xrightarrow{\Delta d} d_{s_3}} \quad \underbrace{\text{France}}_{\xrightarrow{\Delta d} d_{s_4}} \quad \underbrace{\text{is}}_{\xrightarrow{\Delta d} d_{s_5}}$$

In addition to the sentence "The capital of France is", we study the following sentences:

- "The Great Gatsby is my favourite";
- "O Romeo, Romeo, wherefore art thou".

We report our full results for translation in Figures 41 to 43.

## D.2 (LACK OF) SCALE INVARIANCE

On the other hand, for scaling we increase/decrease the duration of the entire sentence:

$$\underbrace{\text{The}}_{d_{s_1} > 1} \underbrace{\text{capital}}_{d_{s_2} > 1} \underbrace{\text{of}}_{d_{s_3} > 1} \underbrace{\text{France}}_{d_{s_4} > 1} \underbrace{\text{is}}_{d_{s_5} > 1}$$

See Figures 44 to 46 for our results.

## D.3 ANALYSIS

As shown for instance in Figure 47, we find that while the impact of shifting on an LLM's output is negligible, scaling significantly changes how the LLM interprets the input (and thus what the LLM outputs). This phenomenon occurs regardless of the model and sentence, empirically confirming the theoretical observations we already made about translation invariance in Section 3.

We believe that shifting does not affect an LLM's output due to the fact that positional and rotary-embeddings (Vaswani et al., 2017; Gemma Team et al., 2024b) are robust to translations: as long as the relative positions between tokens are preserved, a model's output remains consistent. On the other hand, scaling leads to significant variations in an LLM's output.

Our results suggest that beyond interpreting time continuously, duration is itself an intrinsic property of our generalised Transformers that can explain why LLMs are robust on inputs with low frequency in the training data (e.g., their embedding may interpolate with others with similar semantics).

## D.4 SHIFTING INVARIANCE WITH LEARNED POSITIONAL EMBEDDINGS

While the properties of common positional encodings (in particular sinusoidal position encoding and Rotary Positional Encoding) inherently incentivise translation invariance, we study whether such a phenomenon takes place in models with learned positional encodings. To do so, we repeat our experiments with translation in GPT2, which uses this form of encoding.

We report our results in Figure 48. While the magnitude of the effect is certainly less strong compared to RoPE encodings, we observe that the top class remains consistent under translation for moderate shifts, which is consistent with our RoPE results.

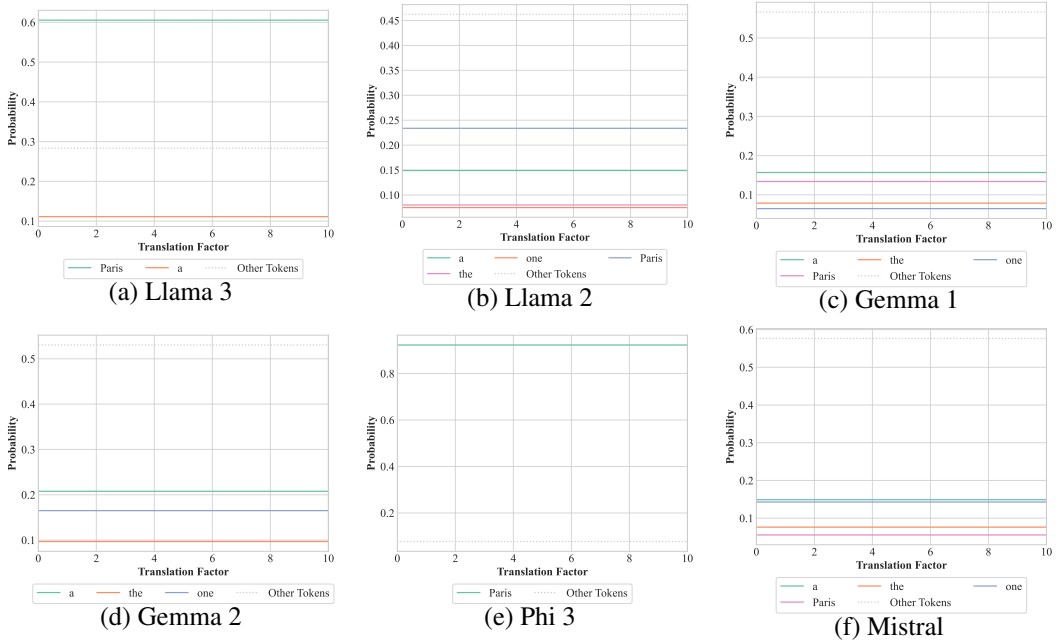

Figure 41: Predicted next token for the sentence "The capital of France is" in all studied models with shifting. We report all tokens with a probability of at least 5% at any point of the interpolation.

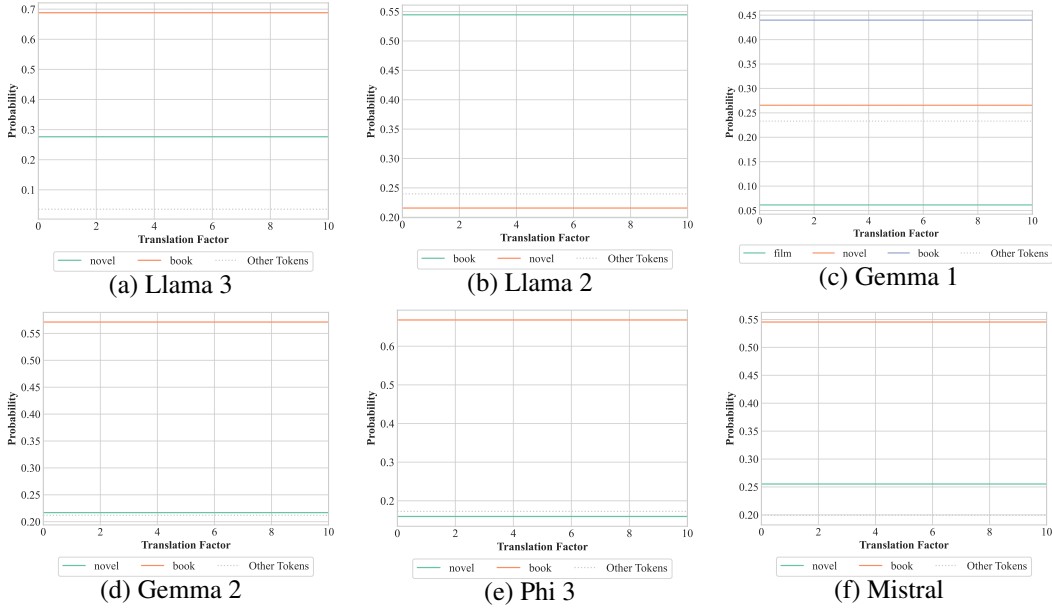

Figure 42: Predicted next token for the sentence "The Great Gatsby is my favourite" in all studied models with shifting. We report all tokens with a probability of at least 5% at any point of the interpolation.

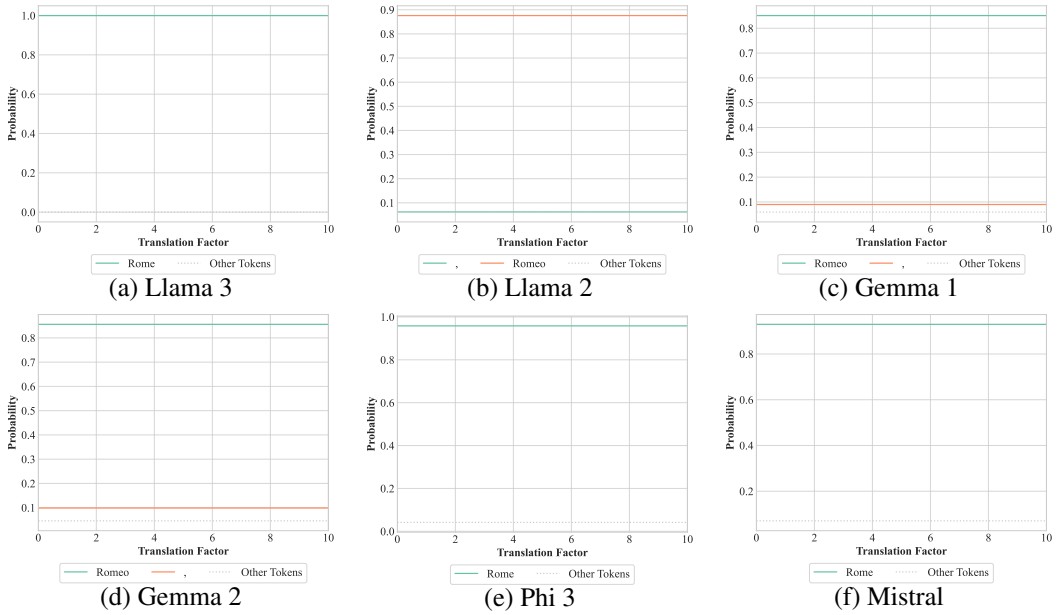

Figure 43: Predicted next token for the sentence "O Romeo, Romeo, wherefore art thou" in all studied models with shifting. We report all tokens with a probability of at least 5% at any point of the interpolation.

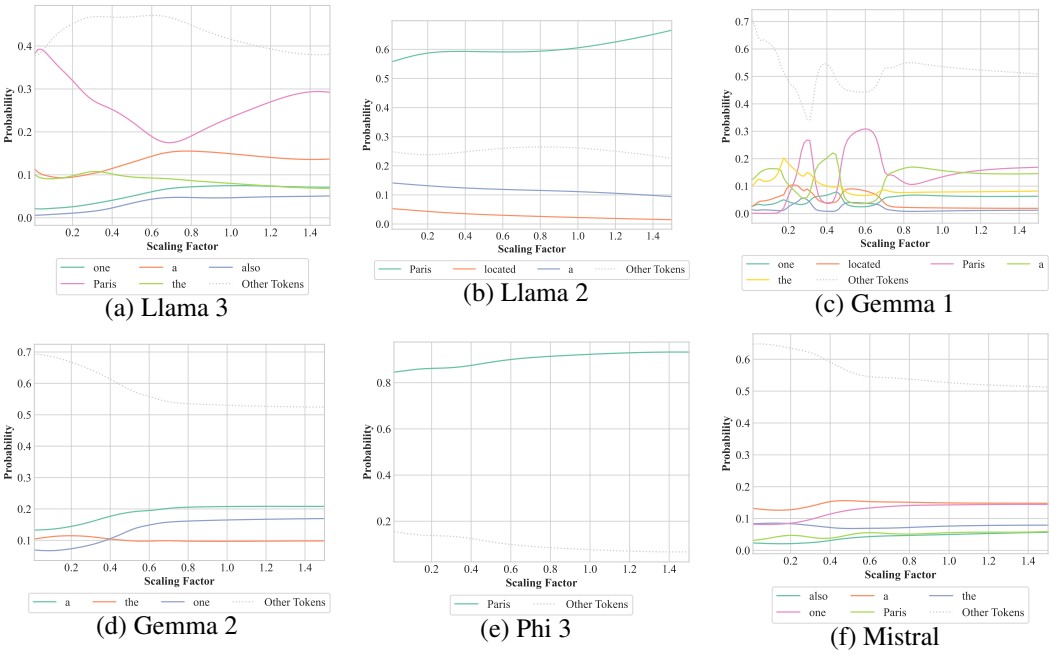

Figure 44: Predicted next token for the sentence "The capital of France is" in all studied models with scaling. We report all tokens with a probability of at least 5% at any point of the interpolation.

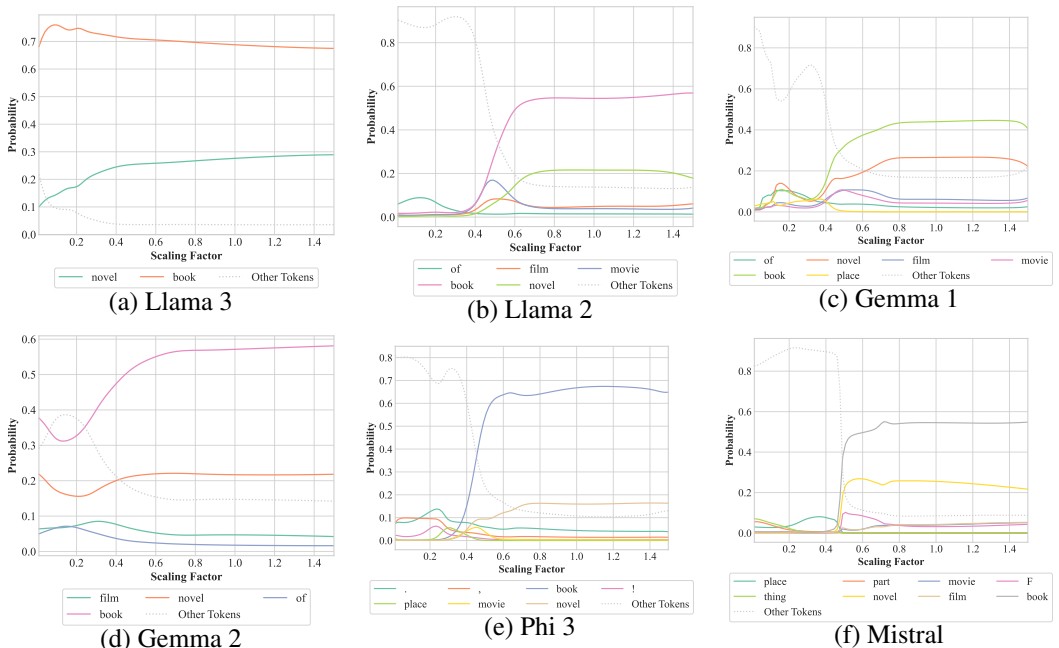

Figure 45: Predicted next token for the sentence "The Great Gatsby is my favourite" in all studied models with scaling. We report all tokens with a probability of at least 5% at any point of the interpolation.

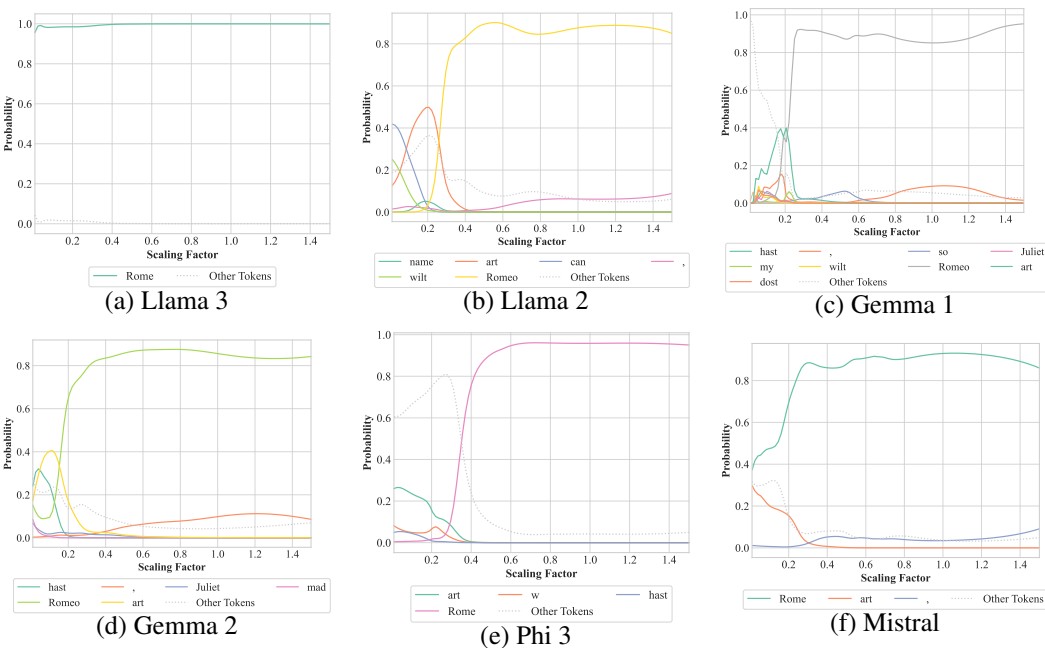

Figure 46: Predicted next token for the sentence "O Romeo, Romeo, wherefore art thou" in all studied models with scaling. We report all tokens with a probability of at least 5% at any point of the interpolation.

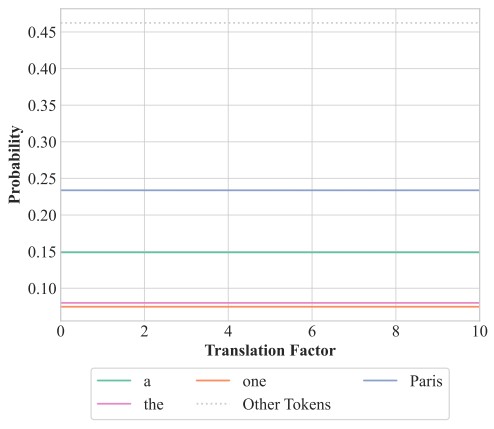

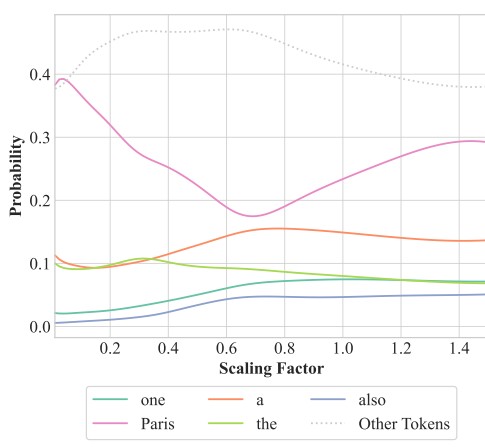

(a) Shifting does not impact an LLM's output.

(b) Scaling visibly impacts an LLM's output.

Figure 47: Effect of shifting and scaling on Llama 3 for the sentence "The capital of France is".

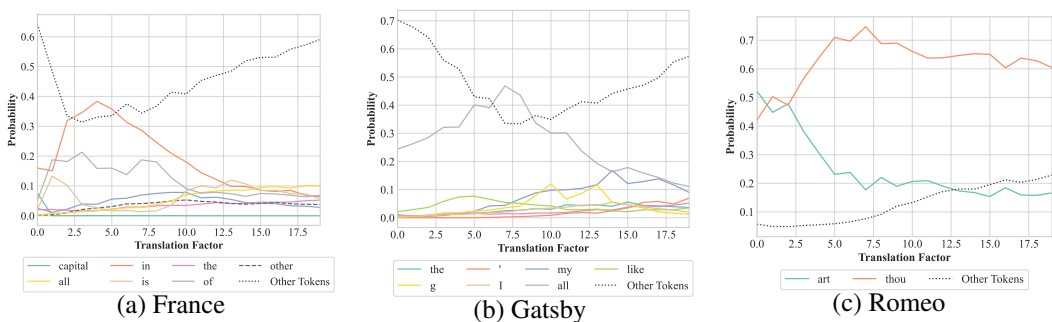

(a) France

(b) Gatsby

(c) Romeo

Figure 48: Predicted next token for the France, Gatsby and Romeo sentences in GPT2 with translation.

