# OpenReview forum: "Language Models Are Implicitly Continuous"
_ICLR.cc/2025/Conference — ICLR 2025 Poster_

### Official Review · Reviewer_hfUi · 2024-10-28

**Soundness:** 2
**Presentation:** 2
**Contribution:** 2
**Rating:** 5
**Confidence:** 3

**Summary:**

The paper looks at transformer architecture (function) from a step function mixture viewpoint and proposes a continuous generalization of the transformer architecture by introducing token duration. From here, the paper investigates into two genres of properties of the transformer architecture (both quite linear) from the transformer architecture:
- time wise, the architecture is shown to be quite linear with time (through for example counting experiments)
- space wise, the architecture is shown to be able to perform linear interpolation on semantic space

The paper also shows some invariance/variance properties of currently used position embeddings.

**Strengths:**

The experiments that vary the duration is interesting and looks novel to me; besides, it follows directly from the established continuous extension of the transformer architecture. The paper shows that the counting or to some extent meaning retention has some correlation with the token duration through various experiments.

It is nice to see semantic interpolation for LLMs, although this is a well known results for word embeddings.

**Weaknesses:**

While the time duration looks interesting and novel, I am not able to tell from the paper what are the experimental settings used to produced the results in the paper (I have put my exact questions in the question section). Thus unfortunately, I am not able to reproduce such experiments and I am not able to see any insights for cause and effects of this phenomenon.

shifting-invariant and scale-variant are well known effects and can be derived directly for models using ROPE embeddings and such properties hold similarly for a large range of models. Such contribution in the paper does not match ICLR acceptance standard, I think.

**Questions:**

What are the exact experimental settings to produce results in 4.1 and 4.2 please? More precisely:
- How are durations taken into account in the modification? Are these done through attention modification as shown in eq (4)?
- How are position embeddings handled in those experiments? When duration is longer/shorter, do authors also adjust position embeddings to reflect the duration change please?

---

> ### Author Response · Authors · 2024-11-21
> **Re: Official Review of Submission11994 by Reviewer hfUi**
>
> We thank the reviewer for pointing out the strengths of our paper. In particular, the reviewers highlighted a contribution that we consider one of the main pillars of our work, i.e., **a theoretical framework that suggests that state-of-the-art LLMs perform semantic interpolation in input and output space**.
>
> The reviewer correctly points out that semantic interpolation is a well-known phenomenon in word embeddings (e.g., Word2Vec).
> Our work extends that analysis by showing that, theoretically and practically, state-of-the-art LLMs embed a **continuous notion of semantics in output space**: to make an analogy, the famous “king-queen ~ man - woman” behaviour of tokens in embedding space is preserved when we prompt a model with a sentence that elicits such analogy (we reported several examples of this behaviour in the Experimental Section of the paper and the Appendix).
> In other words, LLMs treat semantics as a continuous spectrum in the output space. LLMs transform input sentences via a long series of non-linear transformations: the above-mentioned results are thus surprising and shed light on the **structural differences between human and machine language**.
>
> Such behaviours, including shift-invariance and scale-variance, naturally arise from the theory when we extend the Transformers to handle continuous input sequences. We stress that our extensions leave the Transformer and the LLM **intact** in terms of weights, i.e., the LLM behaves **precisely** as in the discrete case when prompted with tokens each of length one.
>
> > shifting-invariant and scale-variant are well known effects and can be derived directly for models using ROPE embeddings and such properties hold similarly for a large range of models. Such contribution in the paper does not match ICLR acceptance standard, I think.
>
> Regarding RoPE and its beneficial effects on shift-invariance and scale-variance, we acknowledge that ROPE injects information that a model can use to make it behave according to the abovementioned principles.
> On the other hand, the magnitude of the beneficial effect of RoPE in this sense is debated: in [1], the authors show that LLMs encode such information without RoPE. As an extreme case, we added a further experiment with GPT-2, an LLM that uses neither RoPE nor the classic sin/cosine positional embedding but instead learns positional embeddings from scratch. As shown for instance in Figure 35 (a), while the variation is higher compared to the RoPE case, the top predicted token for translations of up to 10 tokens is consistent.
>
> This observation, combined with the fact that RoPE also encodes information on the absolute position, enables us to argue that scale-variance and shift-invariance are byproducts that are intrinsic to powerful LLMs and emerge from training rather than implicit biases such as RoPE (though the latter certainly helps by injecting specific inductive biases in the model).
> To conclude, while shift-invariance and scale-variance are additional observations that we consider complementary to the theoretical framework and the space- and time-continuity, we thank the reviewer, and we will add a proper discussion of this phenomenon in the revised version of the paper. We are confident this will make the contribution stronger.
>
> **Questions.**
>
> > How are durations taken into account in the modification? Are these done through attention modification as shown in eq (4)?
>
> Yes, in practice we use custom attention weights, which for a careful choice of weights results in a formulation that is equivalent to the discretisation of Equation 4. We’ve added a new appendix (Appendix B) with the derivation of this property and further experimental setup info.
>
>
> > How are position embeddings handled in those experiments? When duration is longer/shorter, do authors also adjust position embeddings to reflect the duration change please?
>
> Yes, the position embeddings are adjusted as well. For instance, if a token has position embedding $t$ and duration $d$, the following token will have position $t + d$. We have clarified this aspect in the revised paper.

---

> > ### Comment · Reviewer_hfUi · 2024-11-23
> >
> > Thanks for the authors' feedback. The authors acknowledge and address my concerns on the duration explicit modelling and implicit learning aspects. I hope the authors can make it clearer for future readers. I have adjusted my general score from 3 to 5.

---

> > > ### Author Response · Authors · 2024-11-24
> > > **Thank you for the response.**
> > >
> > > We thank the reviewer for participating in the rebuttal and increasing their score.
> > > We will soon add more results to strengthen our analysis, reply to other reviewers' concerns (which may also be of interest to you), and potentially increase further your opinion of our work!

---

> ### Author Response · Authors · 2024-11-21
> **Re: Official Review of Submission11994 by Reviewer hfUi**
>
> References:
>
> [1] Haviv et al., Transformer Language Models without Positional Encodings Still Learn Positional Information.

---

### Official Review · Reviewer_T7Uh · 2024-10-31

**Soundness:** 2
**Presentation:** 3
**Contribution:** 3
**Rating:** 6
**Confidence:** 4

**Summary:**

This paper shows that Transformer language models (LM) implicitly learns continuous sentence representations in both space and time dimensions. The paper generalizes the standard discrete LM inputs to continuous cases, and demonstrate some interesting phenomena like stretching input sequence lengths can bias predictions accordingly (time continuity), and interpolating two inputs enables the model to generate meaningful outputs (space continuity). The paper argues that treating LM as continuous models deepens our understanding of its nature and promises performance improvement.

**Strengths:**

The paper provides a novel perspective of analyzing LMs, that is its representations are implicitly as continuous. This is an interesting and insightful observation that deserves follow-up analysis and further investigation.

**Weaknesses:**

Evidences supporting the claim that LMs are implicitly continuous are not sufficient and limited. Deeper analysis and more diverse experiments are desirable.

**Questions:**

1. If the claim that LMs are temporally continuous, then what if we give them input sequences which are curves that fit the discrete token inputs? In other words, we can consider discrete token inputs as "samples" from continuous signals, so it's natural to ask what would the outputs look like if we input the LM with the original "continuous" signals.

2. Evidences provided by the paper to support the claim that LM is "spatially continuous" (Sec 4.4) are pretty much the same as the classical word vector algebra (e.g. "queen" - "woman" + "man" = "king"), and kind of expected. It might be more interesting to provide some more sophisticated examples that are unique to LM's capabilities, for example interpolating between two logic expressions that a LM can predict well, and observe whether the model is able to find novel logical relations.

3. While it is interesting to provide observations indicating that LMs are time and spatial continuous, what are the implications of this finding on controlling LM behavior and improving its quality? The paper only briefly mentioned some items in Sec 5, but they do not appear to be very concrete or convincing.

4. Beyond the examples that the paper provided, it would be more interesting and insightful to train a LM with inputs being compressed or stretched (like in the time duration cases), or just do inference on a wider variety of more difficult tasks (e.g. reasoning, coding, maths etc.), and compare model behavior differences.

---

> ### Author Response · Authors · 2024-11-21
> **Re: Official Review of Submission11994 by Reviewer T7Uh**
>
> We thank the reviewer for pointing out the strengths of our paper. In particular, the reviewer underlines a contribution that we consider one of the main pillars of our work, i.e., **a theoretical framework that suggests that state-of-the-art LLMs perform semantic interpolation in input and output space**.
>
> >Evidences supporting the claim that LMs are implicitly continuous are not sufficient and limited. Deeper analysis and more diverse experiments are desirable.
>
> Our experiments have been designed to be sufficient and consequential to the theory we developed in Section 3. **We nonetheless agree that more experiments would strengthen our contribution. In fact, we have added experiments to study the translation invariance of models with learned positional embeddings and the interpolations of logical operators** (Appendix D).
> These experiments confirm what we observed in the original submission, and we hope they strengthen the overall reviewer’s opinion of the paper. We are also extending our existing experiments to larger scales and plan to report the results as soon as they are done.
>
> > If the claim that LMs are temporally continuous, then what if we give them input sequences which are curves that fit the discrete token inputs? In other words, we can consider discrete token inputs as "samples" from continuous signals, so it's natural to ask what would the outputs look like if we input the LM with the original "continuous" signals.
>
> Yes, we agree; that is an aspect that we plan to develop in future works. It has also been a very common topic of discussion among us authors! With Transformers that are extended to handle space- and time-continuity, and our results that confirm they behave accordingly to the “continuous” mechanics we discuss in the paper, **the analogy with diffusion models appears evident**. On the other hand, like diffusion models, finding the _right_ continuous process is challenging as the observations one can make are finite, and there are thus infinitely many functions to fit them. It remains, nonetheless a fascinating open problem we plan to study in future.
>
> > Evidences provided by the paper to support the claim that LM is "spatially continuous" (Sec 4.4) are pretty much the same as the classical word vector algebra (e.g. "queen" - "woman" + "man" = "king"). It might be more interesting to provide some more sophisticated examples that are unique to LM's capabilities, for example interpolating between two logic expressions that a LM can predict well, and observe whether the model is able to find novel logical relations.
>
> We have added such an experiment in the revised paper, where we interpolate between logical operators and study the behavior of intermediate representations. We observe the emergence of fuzzy operators whose truth values are best represented as floating points. See for instance Figure 41, which shows the predicted probabilities of AND-OR interpolations in Mistral.
> Note that for truth values that are the same in AND and OR, we see subtle shifts, while for different truth values, we observe fuzzy shifts in output probability.
>
> > While it is interesting to provide observations indicating that LMs are time and spatial continuous, what are the implications of this finding on controlling LM behavior and improving its quality? The paper only briefly mentioned some items in Sec 5, but they do not appear to be very concrete or convincing.
>
> We agree; we mentioned possible research directions and could have expanded on them further. We will add a proper discussion of four research directions we envision.
> One of them is **faster fine-tuning**: our results show that it is possible to feed an LLM with multiple template inputs with some tokens that are the interpolation of multiple tokens. That allows to speed up fine-tuning by processing multiple interpolated inputs simultaneously. One open question is whether it is possible to disentangle the corresponding output and assign it to the correct interpolated input token.
> By extension, one can think of **pre-training an LLM** with the same technique. While some challenges to resolve seem immediately manageable (e.g., one probably needs to scale the output logits when the LLM is fed with sequences that have an interpolation factor different from that at training time), others represent open problems that go beyond the scope of this work (e.g., assigning each predicted token to the correct interpolation to guide the loss function).
> Another direction, more on the linguistics side, **draws a parallel, in terms of similarities and dissimilarities, between human and machine language in the light of our findings**. There are many cases where treating language as continuous leads to paradoxical results (e.g., as per Fig. 2, the 4 “apple” tokens are treated as if the number of occurrences were lower).

---

> > ### Comment · Reviewer_T7Uh · 2024-11-28
> >
> > I'd like to thank authors for the responses they provided to my concerns. I think the additional experiments and the paper revision  plan will improve the clarity and technical depth of the submission. Although the paper could have done better in revealing more interesting properties when a LM is casted as a continuous signal processor, this analysis angle and observations are intriguing and deserves more attention as well as follow-up discussions. I therefore updated my rating to 6.

---

> ### Comment · Area_Chair_rANP · 2024-11-25
> **Rebuttal period ending**
>
> Please engage with the author response. Have they satisfied you as to the significance of their findings and the completeness of their investigation?

---

### Official Review · Reviewer_8hjw · 2024-10-31

**Soundness:** 2
**Presentation:** 3
**Contribution:** 2
**Rating:** 6
**Confidence:** 4

**Summary:**

This paper aims to demonstrate that transformer-based language models not only learns to process input sequences consisting of discrete tokens, but are also able to "reason" about the continuous interpolation of discrete tokens along the temporal and spatial dimensions. The paper first presents a continuous generalization of the classical transformer model to make it accept spatially and temporally interpolated continous inputs. It then shows via exemplar analyses that one can manipulate the "temporal duration" of input tokens to control model prediction on simple reasoning tasks; it also shows that the "spatial interpolation" of input tokens, formulated as their weighted averages, also elicit meaning outputs from the tested language models. The authors therefore claim that LLMs process natural language in a way that is fundamentally different from human speakers.

**Strengths:**

1. The continuous generalizations of key transformer modules such as multi-head self-attention seem novel and theoretically sound.

2. The presented examples of controlled input manipulation and the resulting model predictions in section 4 are interesting and intuitively make sense.

**Weaknesses:**

1. Despite the novelty of the main arguments and the proposed continuous generalization of transformers, the paper did not provide suffcient empirical evidence and quantitative evaluation to support them. In particular, the analyses in section 4 are based on only a couple of example input sequences, with a very special type of prompt template. It would make this section more convincing if the authors could curate and evaluate LLMs on a dataset of "temporally" or "spatially" interpolated input sentences, and define a more general evaluation metric to capture the variation of model prediction when manipulating the time durations or spatial weights of target tokens.


2. The claim that "LLMs reason about language (continuously)" seems also to be an overstatement given the input- and output-level simplicity of the evaluation tasks (e.g. counting the mentions of certain entity type in a short sentence). I'm curious to see whether the model could still produce fluent outputs on more complex text generation tasks (e.g. long-form QA) if we manipulate input tokens as suggested in section 3. If not, it would suggest that LLMs in fact do not assign meaningful representations for the spatio-temporally interpolated counterparts of input token embeddings.

**Questions:**

For the single-token time continuity test in section 4.1, what would happen if you use input prompts with different entity names from the same category? (e.g. In the sentence "apple banana pear orange", how many fruits are mentioned?)

---

> ### Author Response · Authors · 2024-11-21
> **Re: Official Review of Submission11994 by Reviewer 8hjw**
>
> We thank the reviewer for pointing out the strengths of our paper. In particular, a contribution we consider one of the main pillars of our work, i.e., **a theoretical framework that suggests that state-of-the-art LLMs perform semantic interpolation both in input and output space**.
> We also thank them for **appreciating the experimental section**.
>
>
> > the paper did not provide suffcient empirical evidence and quantitative evaluation to support them. In particular, the analyses in section 4 are based on only a couple of example input sequences, with a very special type of prompt template. It would make this section more convincing if the authors could curate and evaluate LLMs on a dataset of "temporally" or "spatially" interpolated input sentences, and define a more general evaluation metric to capture the variation of model prediction when manipulating the time durations or spatial weights of target tokens.
>
> We are currently running a batch of experiments to give further evidence of time- and space-continuity in state-of-the-art LLMs. **We will update this thread as soon as we gather the results and metrics.**
> As regards the introduction of new metrics, we will add them in the finalised version of the paper, and here we expand a bit on how we will define them.
>
> For instance, before Section 3.2, we will introduce **time-sensitivity** as the **weighted average of the time duration of each token the model outputs** (i.e., that with max probability) during the manipulation. Consider the example in Figure 2. In that case, the time-sensitivity would be 1., as the model shows peaks for four different output tokens, and the input contains four times the word ‘apple’. In other words, a simple measure would be $k/n$, where $$k is the number of different tokens the model outputs and n is the total number of tokens of which we manipulate their time duration.
> In the same spirit, we will introduce a notion of **space-sensitivity**.
>
> In the final version of the paper, **all the results will account for similarly quantitative metrics**.

---

> ### Comment · Reviewer_8hjw · 2024-11-21
>
> I appreciate the detailed responses by the authors, and I look forward to their additional experimental results to support the main theoretical contribution of this paper. I'll adjust my score based on the soundness of the new results.

---

> > ### Author Response · Authors · 2024-11-21
> >
> > Thank you, we are looking forward to presenting them as soon as possible.
> >
> > (And thank you for the correction; we have deleted the reply and moved it under the right thread).

---

> ### Author Response · Authors · 2024-11-25
> **Quantitative Results**
>
> Dear Reviewer 8hjw,
>
> We have just completed the analysis of our quantitative results for time sensitivity. We used the sequential dataset from [1], which contains 200 curated how-to tutorials, divided into steps.
> Here is an example element from the dataset (formatted for our specific use case):
> ```
> Tutorial: How to Remove Gorilla Glue from Wood
> - Use a paint scraper to remove most of the glue.
> - Use a chisel to remove smaller amounts of glue.
> - Sand off the remaining glue.
>
> Question: How many steps are necessary to complete the tutorial? Reply with a single-digit number
> Answer: It takes
> ```
>
> We reduce the duration of all the steps (following the procedure described in Appendix B) and measure the time sensitivity, i.e. the number $k$ of unique applicable token peaks divided by the expected number of peaks $n$. For instance, if there are 4 steps, we expect to see peaks for 1, 2, 3 and 4*.
>
> We define a unique token peak as any token $x_t$ such that there exists a duration factor $\varphi \in [0.1, 1]$ for which $$P(x_t | x_0, \dots, x_{m-1}, s(x_m \dots x_{q};\varphi), x_{q+1}, \dots x_{t-1}) \geq \tau,$$ where $\tau$ is a peak threshold and $s(x_m \dots x_{q};\varphi)$ is the portion of the tokens with index in $[m, q]$ whose duration is reduced by a factor of $\varphi$. In our context, the indices $[m, q]$ are the indices of the tokens representing the tutorial steps.
>
> A discrete interpretation of LLMs would predict that the model consistently assigns the same probabilities to the same tokens regardless of the duration factor. This would correspond to the time sensitivity computed with a fixed $\varphi = 1$ (i.e. no shrinking). We thus compute the expected average time sensitivity for the sequential dataset assuming that, instead of our hypothesis, the discrete interpretation is correct. We name such measure the _counterfactual time sensitivity_.
>
> Our results are reported in Figure 48.
>
> For very high thresholds $\varphi$, the time sensitivity and the counterfactual time sensitivity are roughly the same, since nothing is counted as a peak and thus both scores are close to 0. However, for all other values, the time sensitivity is significantly higher than the counterfactual time sensitivity.
>
> We report below the AUCs of the time sensitivity and counterfactual time sensitivity, which further confirm that time sensitivity is higher than what we would expect should the discrete interpretation be correct:
>
> | Model | Counterfactual Time Sensitivity | Observed Time Sensitivity |
> |---|---|---|
> | Llama 3 8b | 0.1316 | 0.2746 |
> | Llama 2 13b | 0.1507 | 0.4412 |
> | Gemma 7b | 0.1349 | 0.2453 |
> | Gemma 2 9b | 0.1581 | 0.3014 |
> | Phi 3 | 0.1660 | 0.3782 |
> | Mistral | 0.1429 | 0.2297 |
>
>
> In other words, **our quantitative results are incompatible with a discrete interpretation of LLMs**, which is further evidence in support of our thesis. See Table 1 in the paper for a tabular visualisation of these results.
>
>
> Note that, in our analysis, we only measure the number of “correct” peaks divided by the number of expected peaks: for instance, if there are four steps and we observe peaks for 1, 2, 3, 4 and 5, our time sensitivity score is 1, even if there is a spurious peak (namely 5). While the presence of extra peaks does not contradict our thesis that LLMs meaningfully respond to variations in duration, for the sake of thoroughness we also compute another metric, namely the Intersection over Union, defined as follows:
> $$
> \frac{size(\text{uniquePeaks} \cap \text{expectedPeaks})}{size(\text{uniquePeaks} \cup \text{expectedPeaks})}
> $$
>
> Our IoU scores are reported in Figure 49.
>
> Thank you again for your feedback and we hope that such quantitative results provide further evidence of the incompleteness of the discrete interpretation of LLMs.
>
>
>
> [1] Lin, Fangru, et al. "Graph-enhanced Large Language Models in Asynchronous Plan Reasoning." ICML 2024.
>
> *We treat 0 as an unexpected peak due to the fact that the semantics of a zero-duration sentence are ill-defined.

---

> > ### Comment · Reviewer_8hjw · 2024-11-25
> > **Thank you for the additional experiments and results.**
> >
> > I appreciate the additional experimental results that the authors report in their rebuttal, and I think they have made the main argument about LLMs' time-continuous computation more convincing. I have therefore raised my score to 6, and I would suggest the authors to put the new results (and presumably some discussions) in the main text to improve the soundness of their framework.

---

> > > ### Author Response · Authors · 2024-11-26
> > > **Response**
> > >
> > > We thank the reviewer for engaging in the discussion and being open to improving their point of view on our work.
> > > We will add the results and the discussion in the final version of the paper to make a more substantial contribution to the topic.

---

### Official Review · Reviewer_R12D · 2024-11-02

**Soundness:** 3
**Presentation:** 3
**Contribution:** 3
**Rating:** 6
**Confidence:** 4

**Summary:**

In this paper, the authors explore how Transformer-based language models, which traditionally operate on discrete token sequences, can implicitly represent language in a continuous manner. They introduce a novel continuous extension of Transformers, termed the Continuous Causal Transformer (CCT), allowing models to handle continuous inputs in both time and space. Through experiments on state-of-the-art models like Llama3 and Mistral, they demonstrate that these models exhibit continuity in their internal representations, causing different responses to variations in input duration and interpolation between tokens. The findings suggest that these models create implicit, continuous representations that differ fundamentally from human linguistic understanding.

**Strengths:**

The paper explores the intriguing hypothesis that language models, despite being trained on discrete sequences, learn to form inherently continuous representations of language. To investigate this, the authors propose a formalization of continuous Transformers, which "collapse" back to the discrete case when evaluated at discrete time steps. This formulation offers flexibility to probe the nature of model representations, such as by adjusting the "duration" of tokens in a sequence. The paper clearly articulates this formulation and its expressive power, laying a solid foundation for the subsequent experiments.

The experiments logically build on this framework and provide support for the hypothesis that models represent language in a continuous manner. For example, by modifying the duration of tokens in a sequence of repeated items, the model produces varied answers to simple counting tasks, indicating a sensitivity to continuous input features. In another set of experiments, interpolation between the representations of two tokens shows how the model continuously change its prediction according to the "weight" of each token in the interpolation.

Overall, the main contribution of the paper is providing a framework for thinking about LMs as continuous processes, which lends itself to simple probing experiments on the nature of LM representations.

**Weaknesses:**

I don't find any weakness in the theoretical formulation. I have some reservations regarding the *interpretations* of the findings and the lack of reference to previous work on the subject:

* First, I am not entirely clear on the details of the experiment in Section 4.1. Am I correct in understanding that this setup is no longer equivalent to a "regular," discrete Transformer, as the duration of different tokens is no longer uniform? Could the model, in principle, have learned a different behavior (i.e., the "human" interpretation of the sequence)? Specifically, is it correct that when using, for instance, a duration of 0.25 for each of the four occurrences of "apple," the attention dedicated to them would be identical to the attention given to a single occurrence of "apple" in a standard Transformer? If this is indeed the case, the result is unsurprising, as the representations fed into the model would be identical to those in the prompt, "In the sentence 'apple,' how many fruits are mentioned?". In that case, the manipulation in that experiment is just a different way to acknowledge the fact that LMs are fed continuous vectors and not discrete symbols.

* The interpolation experiments in Section 4.1 do provide evidence of the continuous nature of language model representations. However, these findings are not novel, and the paper lacks sufficient references to a substantial body of prior work. The continuous representation of distinct symbolic linguistic processes dates back to the creation of dense vectors via SVD of PMI matrices. Additionally, in interpretability research, there has been extensive work on the continuous representation of symbolic properties, such as linear concept erasure, which identifies specific manipulations in the latent space to isolate particular concepts.

**Questions:**

One possible use case of the theoretical framework proposed in the paper is probing which continuous space model best matches the one learned by the LM, i.e., we can come up with different continuations schemes and examine which best matches the observed one. Does that make sense? do you have any thoughts about that?

---

> ### Author Response · Authors · 2024-11-21
> **Re: Official Review of Submission11994 by Reviewer R12D**
>
> We thank the reviewer for pointing out the strengths of our paper. In particular, what we consider one of the main pillars of our work, i.e., **a theoretical framework that suggests that state-of-the-art LLMs perform semantic interpolation both in input and output space**.
> We also thank them for appreciating the experimental section.
>
> > Am I correct in understanding that this setup is no longer equivalent to a "regular," discrete Transformer, as the duration of different tokens is no longer uniform? Could the model, in principle, have learned a different behavior (i.e., the "human" interpretation of the sequence)?
>
> No, our extensions leave the Transformer and the LLM **intact** in terms of weights and activations, i.e., the LLM behaves **precisely** as in the discrete case when prompted with tokens each of length one. In other words, our extension works precisely as the discrete case for uniform token durations and each of length one.
>
> > Specifically, is it correct that when using, for instance, a duration of 0.25 for each of the four occurrences of "apple," the attention dedicated to them would be identical to the attention given to a single occurrence of "apple" in a standard Transformer?
> No, that is incorrect: generating the output for each “apple” token after the first one involves attending the previous “apple” tokens, resulting in a different output sequence. Moreover, each apple token has a different positional embedding (e.g. $t$ for the first “apple” token, $t + 0.25$ for the second, $t + 0.5$ for the third, $t + 0.75$ for the fourth, and $t + 1$ for the following token).
>
> In that case, the manipulation in that experiment is just a different way to acknowledge the fact that LMs are fed continuous vectors and not discrete symbols.
> We agree that our work theoretically frames the math to build continuous Transformers, i.e., Transformers that handle (continuous) functions.
> On the other hand, we observe that this extension, for which the Transformers behave for inputs of canonical length, precisely as in the discrete case, evidences how these models embed a continuous notion of semantics in input and output space. These observations force us to rethink how LLMs treat language, as it is hard to reconcile with the standard theories on language (i.e., phonemes in linguistics are inherently discrete, and there are nearly no attempts at modelling them as continuous). At the end of the paper, we provide some intriguing open problems to explore as future works, including cutting the cost of LLMs pre-training with input templates and interpolation.
>
> > However, these findings are not novel, and the paper lacks sufficient references to a substantial body of prior work. The continuous representation of distinct symbolic linguistic processes dates back to the creation of dense vectors via SVD of PMI matrices.
>
> The reviewer correctly points out that semantic interpolation is a well-known phenomenon in word embeddings (e.g., connecting Word2Vec with PMI, SVD, etc.).
> Our work extends that corpus by showing that, theoretically and practically, state-of-the-art LLMs embed a **continuous notion of semantics in output space**.
> To make an analogy, the famous “king-queen ≈ man-woman” linear behaviour observed in embedding space is **preserved in output space** when we prompt a model with a sentence that elicits such an analogy. We reported several examples of this behaviour in the Experimental Section of the paper and the Appendix.
> In other words, LLMs treat semantics as a continuous spectrum in the output space. LLMs transform input sentences via a long series of non-linear transformations: the above-mentioned results are novel and shed light on the **structural differences between human and machine language**.
>
> > Additionally, in interpretability research, there has been extensive work on the continuous representation of symbolic properties, such as linear concept erasure, which identifies specific manipulations in the latent space to isolate particular concepts.
>
> We thank the reviewer for pointing out works in interpretability that look at the problem of erasing concepts in embedding space. Our work is complementary in the sense that it can be used to theoretically and practically find regions in the embedding space associated with a particular concept and eventually erase them. Without acknowledging and formally proving that LLMs treat semantic concepts as a continuum, erasure cannot provide guarantees if applied to single vectors.

---

> > ### Author Response · Authors · 2024-11-21
> >
> > > One possible use case of the theoretical framework proposed in the paper is probing which continuous space model best matches the one learned by the LM, i.e., we can come up with different continuations schemes and examine which best matches the observed one. Does that make sense? do you have any thoughts about that?
> >
> > Yes, we agree; that is an aspect we plan to develop in future work. (we discussed it among the authors multiple times!) With Transformers extended to handle space- and time-continuity, and our results that confirm they behave accordingly to the “continuous” mechanics we discuss in the paper, **the analogy with diffusion models appears evident**. On the other hand, similar to diffusion models, finding the _right_ continuous underlying process is challenging as the observations one can make are finite, with infinitely many functions that fit them. It remains, nonetheless a fascinating open problem we plan to study in future.

---

> > ### Comment · Reviewer_R12D · 2024-11-24
> > **Response**
> >
> > Thank you for your details response and explanation, it clarified several points that were initially unclear to me.
> >
> > Please note that previous work has already shown that LMs build meaningful continuous representations. This has been used, e.g., in the words on "linear steering" of LMs by applying affine transformations on their hidden representations in order to affect their behavior.

---

> > > ### Comment · Reviewer_R12D · 2024-11-24
> > > **Response**
> > >
> > > Thanks!

---

> > > > ### Author Response · Authors · 2024-11-25
> > > > **Comparison with Past Work**
> > > >
> > > > Thank you for the response!
> > > >
> > > > We agree that past work has studied continuous representations in LLMs; however, this has been either in the context of concepts for which intuitive spectra exist [1] or from a neuron-driven perspective [2]. In particular, our work can be seen as complementary to the latter: while the authors of [2] show that certain neurons map to specific concepts (which can then be steered), we show that such concepts exist even at the embedding level, and we offer a theoretical framework to study formally such phenomena in an architecture-independent way.
> > > >
> > > > Additionally, our work can also be seen as a response to [3], which trains inherently continuous embeddings: we show that this process is not necessary, as even discretely trained LLMs show similar behaviours.
> > > >
> > > > We have expanded the Related Work section to include a proper comparison with such works. Thank you again!
> > > >
> > > > The Authors
> > > >
> > > > ---
> > > >
> > > > [1] Gurnee, Wes, and Max Tegmark. "Language models represent space and time." arXiv preprint arXiv:2310.02207 (2023).
> > > >
> > > > [2] Anthropic. "Scaling Monosemanticity: Extracting Interpretable Features from Claude 3 Sonnet". Transformer Circuits.
> > > >
> > > > [3] Vilnis, Luke, and Andrew McCallum. "Word representations via gaussian embedding." arXiv preprint arXiv:1412.6623 (2014).

---

> ### Comment · Reviewer_R12D · 2024-11-27
> **Response**
>
> That is not true. Many works point out to the existence of interpretable continuous directions in the representation space of LMs, beyonf neurons. E.g. [1][2][3].
>
> [1] Arditi, Andy, et al. "Refusal in language models is mediated by a single direction." arXiv preprint arXiv:2406.11717 (2024).
>
> [2] Ravfogel, Shauli, et al. "Linear adversarial concept erasure." International Conference on Machine Learning. PMLR, 2022.
>
> [3] Todd, Eric, et al. "Function Vectors in Large Language Models." International Conference on Learning Representations. ICLR, 2024.

---

> > ### Author Response · Authors · 2024-11-30
> >
> > Thank you for the updated list of references. We will add them all to the finalised version of the paper, with a proper discussion of the aspects already studied in the literature and what our theoretical contribution introduces.
> >
> > In particular, in [1] the authors focus on erasing the concept of "refusal" (i.e., a model does not respond to a potentially harmful prompt). While relevant to the discussion, our work shows that continuous behaviours are not limited to concepts but also words (please see the example of four "Apples" in a prompt that are interpreted as a continuous entity when modulating its time duration): we stress that this phenomenon diverges from how humans interpret language, and requires us to be careful when we draw analogies between the language LLMs and humans learn.
> >
> > In [2], the authors remove concepts like biases by erasing the pertinent area that makes that concept emerge. Our work, despite overlapping (we will cite it properly), shows that LLMs can interpolate between "overlapping" concepts: erasing an entire area would also affect other concepts. This represents an interesting research direction one can take to develop our work further.
> >
> > In [3], the authors show, via what they name function vectors, that LLMs can carry references of tasks (in the context of ICL). Related to our work, they show that function vectors are quasi-closed to some operations such as sum. We want to investigate further the connections of our work to this one, thus we will cite it properly. It is possible that some of the continuous behaviours in an LLM arise as function vectors.
> >
> > To conclude, we stress that our theoretical framework complements the references reported, as it allows us to study an LLM's continuous behaviour (i.e., the phenomena mentioned by [1, 2, 3]) and other phenomena seamlessly, i.e., with minimal intervention in the model and by leaving its discrete capabilities intact. Again, thank you for providing these references.
> >
> > The Authors

---

### Author Response · Authors · 2024-11-21

Dear Reviewers,

Thank you for reviewing our article and providing useful feedback that we will use to improve our final version of the paper.

In particular, we are glad all the reviewers acknowledged as novel and valuable what we consider, in order of importance, the three pillars of our work, i.e., 1) **a theoretical framework** that suggests that **state-of-the-art LLMs perform semantic interpolation both in input and output space**, 2) an analysis of how **our extended Transformers can be seamlessly used to analyse their continuous behaviour**, and 3) a series of experiments that show how **LLMs treat inputs as continuous functions, with large implications for how they interpret the semantics of a text**.

Regarding the weaknesses, two reviewers pointed out that the “semantic continuum” where LLMs “reason” was well known since Word2Vec. While we agree that is known for embeddings in input space, **our work shows that state-of-the-art LLMs preserve these analogies in output space**.
We were also asked to clarify the experimental settings: we apologise for not making them very clear and straightforward in the paper; we reported in the revised version the **exact implementation details**. We also added a notebook in the code to **replicate the paper's main results**.

Furthermore, two reviewers asked us to provide more in-depth experiments on the phenomena we described and observed in the paper. In this rebuttal, **we included further experiments**, particularly on translation invariance beyond RoPE and semantic interpolation of Boolean formulas, that confirm what we reported in the Experimental Section of the paper. We are also conducting further experiments, which will be reported in a few days.

We replied to each author individually, but please feel free to read the other reviewers’ comments to have a clearer idea of each reviewer's main strengths and concerns about the paper.

Thank you,

The Authors.

---

### Author Response · Authors · 2024-12-04
**Concluding Thoughts**

Dear Reviewers and Area Chairs,

Thank you for this very productive discussion period. Thanks to your feedback, we have addressed our work's limitations and modified the paper to present our results in the clearest way possible. In particular:
- We have added quantitative results that show that the time-sensitivity of LLMs is incompatible with a discrete interpretation of LLMs, which further proves our thesis;
- We have added experiments showing that LLMs interpolate between binary operators, which represents another example of how LLMs assign non-trivial semantic meaning to intermediate embeddings;
- We have clarified the implementation details of our experiments by adding a new appendix that explains in-depth the mathematics of our HuggingFace-based models;
- We have added references to more works that study the behaviour of LLMs.

We’re also glad that the majority of reviewers significantly increased their scores, which gives us confidence in the importance of our findings.

Once again, thank you for your insight and we hope that our new results have further shown the significance of our theoretical framework & practical results.

Best,

The Authors

---

### Public Comment · ~Samuele_Marro1 · 2025-02-25
**Summary of Camera-Ready Improvements**

Dear Reviewers and Area Chairs,

Thank you again for recommending the acceptance of our paper. Here is a summary of the improvements we have made for the camera-ready version (on top of the improvements for the rebuttal revision):
- We have added quantitative experiments to expand upon the sum-based qualitative results;
- We have replaced our initial quantitative time-sensitivity experiments with a more mathematically grounded analysis based on relative peaks (for the sake of completeness, our time sensitivity results are still available in the Github repo);
- We have added quantitative experiments to study the embedding interpolations in the output space;
- We have moved the translation & scale invariance experiments to give more room to the main results;
- We have added the references suggested by Reviewer R12D.
- We have reordered the appendices to improve readability;
- We have shrunk most appendix images to reduce page count and improve readability

Looking forward to presenting!

Best,

The Authors

---

### Meta-Review · Area_Chair_rANP · 2024-12-19

**Metareview:**

This paper demonstrates that the discrete inputs of language models can be extrapolated or interpolated in continuous ways, making them fundamentally different from how humans process language as a discrete sequence.

Pros:
Conceptually interesting argument that the continuous nature of language models makes them fundamentally different in their language processing from humans.
Convincing demonstration that we can extrapolate the length of a sequence continually and meaningfully interpolate between tokens.

Cons:
Some findings in the paper are reproduced or obvious from existing work.

**Additional Comments On Reviewer Discussion:**

The authors added experiments showing interpolation between binary operators and showing that the language models cannot be operating in a discrete space because of their time sensitivity. They made some presentation improvements, including citing important related work.

---

### Decision · Program_Chairs · 2025-01-22

Accept (Poster)